# 60 cm² perovskite-silicon tandem solar cells with an efficiency of 28.9% by homogeneous passivation

Kerem Artuk [1] ✉, Aleksandra Oranskaia[2], Deniz Turkay[1], Felipe Saenz [3], Mounir D. Mensi[4], Michele De Bastiani [3], Andrés-Felipe Castro-Méndez [5], Julien Hurni [1], Christophe Allebé[3], Mostafa Othman [1], Lisa Champault[3], Austin G. Kuba[1], Alexandra Levtchenko [6], Daniel A. Jacobs[3], Jean-Baptiste Puel[6,7], Daniel Ory [6,7], Felix Lang[5], Aïcha Hessler-Wyser [1], Udo Schwingenschlogl [2], Quentin Jeangros [3], Christophe Ballif [1,3] & Christian M. Wolff [1] ✉

Inverted perovskite solar cells face performance limitations due to non-radiative recombination at the perovskite surfaces in devices, including functional layers. Advanced characterization and density functional theory reveal that phosphonic acids passivate perovskite surface defects, while piperazinium chloride mitigates interface recombination by improving energy level alignment, introducing a field effect, and homogenizing the surface. Together, the quasi-Fermi level splitting of the perovskite is homogeneously increased by ca. 100 mV. This enables two-terminal perovskite-on-silicon tandems to achieve a certified open-circuit voltage of 2 V for a 1 cm² device and high performance in excess of 31%. The scalability of the passivation is furthermore demonstrated with homogeneously passivated devices reaching certified efficiencies of 28.9% for an active area of 60 cm².

In inverted (p-i-n) perovskite solar cells (PSCs), the electron transport layer (ETLs) is typically composed of fullerene ($C_{60}$) or its derivatives[1,2]. These layers effectively extract electrons, and $C_{60}$ has proper band alignment with medium bandgap (~1.52 eV) Pb-based perovskite absorbers, enabling p-i-n devices to achieve certified power conversion efficiency (PCE) beyond 26%[3]. However, a significant drawback arises from the defective interface between the wide bandgap (>1.63 eV) perovskite absorbers and fullerenes, leading to increased non-radiative recombination[4]. Additionally, the wide bandgap perovskite absorber itself poses challenges due to bulk and surface defects[5]. Furthermore, wide bandgap absorbers suffer from a higher

conduction band minima ($E_{CBM}$) offset to the lowest-unoccupied-molecular-orbital (LUMO) of $C_{60}$[6], enhancing Shockley–Read–Hall (SRH)-like interface recombination at this critical interface, leading to high surface recombination velocities (SRVs), reducing the quasi-Fermi level splitting and consequently device-$V_{OC}$[7,8].

Passivation of surface defects (e.g., vacancies, interstitials, substitutional defects, or metallic Pb⁰)[9,10] of Pb-based perovskite absorbers (surface passivation) has been demonstrated using various compounds, including organohalides (oleylamonium iodide[11], butylammonium iodide, 3-methylthiopropylammonium iodide (3-MTPAI)[12], organometals (FcTc₂)[13,14], zwitterions (phenformin hydrochloride)[15], and inorganic

[1]École Polytechnique Fédérale de Lausanne (EPFL), Institute of Electrical and Microengineering (IEM), Photovoltaics and Thin-Film Electronics Laboratory (PV-Lab), Rue de la Maladière 71b, 2000 Neuchâtel, Switzerland. [2]King Abdullah University of Science and Technology, Thuwal 23955-6900, Saudi Arabia. [3]Centre d'Electronique et de Microtechnique (CSEM), Rue Jaquet-Droz 1, 2000 Neuchâtel, Switzerland. [4]École Polytechnique Fédérale de Lausanne (EPFL), Institute of Chemical Sciences and Engineering (ISIC), Rue de L'Industrie 17, 1951 Sion, Switzerland. [5]Physik und Optoelektronik weicher Materie, Institut für Physik und Astronomie, Universität Potsdam, Karl-Liebknecht-Straße 24-25, 14476 Potsdam, Germany. [6]IPVF, 18 Boulevard Thomas Gobert, 91120 Palaiseau, France. [7]EDF R&D, 18 Boulevard Thomas Gobert, 91120 Palaiseau, France. ✉e-mail: kerem.artuk@epfl.ch; christian.wolff@epfl.ch

halides (LiF)[16]. To suppress recombination between perovskite and $C_{60}$ (interface passivation), thin evaporated dielectric interlayers (LiF, $MgF_x$)[17,18], ALD-deposited $AlO_x$[19], and solution-processed organohalides (diammonium salts like propane-diammonium iodide ($PDAI_2$)[20], ethylenediammonium iodide and piperazinium salts like PI[21,22]) and phosphonic acids (PAs) like pentafluorobenzylphosphonic acid (pFBPA) have been utilized effectively[23,24]. To fully exploit the potential of inverted perovskite solar cells, simultaneously suppressing interface and surface recombination holds significant promise. A bimolecular approach can be employed where two molecules with complementary passivation abilities are combined at the surface/interface. This approach has been recently demonstrated by ref. 12. in medium bandgap perovskite absorbers, where surface passivation with 3-MTPAI increased the photoluminescence quantum yield (PLQY) beyond 10% while interface recombination was suppressed by a diammonium halide ($PDAI_2$), resulting in a certified single-junction efficiency of 25.1%. In their later work, combining a bimolecular approach at the interface with additive engineering to minimize bulk defects, ref. 25 achieved a certified efficiency of 26.1%, one of the highest efficiencies for single-junction perovskite solar cells[25]. Likewise, Ugur et al. combined additive engineering using a diammonium chloride with the shortest alkyl chain (methylenediammonium chloride) to minimize bulk defects and $LiF/PDAI_2$ bilayer at the perovskite/$C_{60}$ interface. A certified perovskite-silicon tandem efficiency of 33.7% could be demonstrated with this approach[26]. Such a dielectric/diammonium halide bilayer ($LiF/EDAI_2$) was also shown by Liu et al., who recently achieved a certified tandem performance of 33.89% with an open-circuit voltage of 1.97 V[27]. An effective combination of two thin films at an absorber/contact interface has also been used in the crystalline (c-)Si industry. In passivated emitter and rear contact (PERC) and tunnel oxide passivating contact (TOPCon) solar cells, the $AlO_x/SiN_x$ stack provides a high-quality interface (low SRV) with the combination of field effect (via fixed negative charges in $AlO_x$) and chemical passivation (via dangling bond passivation and hydrogenation by the $AlO_x/SiN_x$ stack)[28,29].

Mitigating voltage losses at the perovskite/$C_{60}$ interface is imperative to exploit the full potential of wide bandgap perovskite absorbers. These strategies are crucial not only for enhancing the performance of individual inverted PSCs but also for tandem/multi-junction devices incorporating inverted perovskite sub-cells. In this work, we combine two molecules (a phosphonic acid, and a piperazinium halide) with complementary passivation abilities. The former passivates perovskite top surface defects, and the latter improves band alignment, introduces a field effect, homogenizes the surface potential overall enables reduced losses when in contact with the electron transport layer.

## Results

### Surface passivation with phosphonic acids

We begin by attempting to passivate top surface defects with PAs (Fig. 1a) via a post-deposition treatment of the herein used triple halide triple cation[30] perovskite absorbers. For that purpose, we fabricate half-cell test structures (ITO/(4-(3,6-dimethyl-9H-carbazol-9-yl)butyl) phosphonic acid(Me-4PACz)/$SiO_x$-np/perovskite) to track the changes in PLQY upon deposition of PAs (Fig. 1a) atop the perovskite. Then, we extract the non-radiative (NR) losses from PLQY measurements (measured from absolute steady-state PL) following: $\Delta V_{OC, non-rad.} = \frac{k_B T}{q} * \ln(PLQY)$. The average NR-loss of the bare absorber is 107 mV (on the hole transport layer stack) at carrier densities of ca. 1 sun (Fig. 1b). Upon deposition of methylphosphonic acid (MPA), the NR-loss increases slightly to an average of 109 mV, decreases slightly to 105 mV with benzyl PA (BPA), with fluorobenzyl PA (FBPA), it decreases to 93 mV and with pentafluorobenzyl PA (pFBPA) it decreases to 85 mV. The decrease in the non-radiative loss with pFBPA deposition shows that either the surface defect density is reduced (e.g., chemical passivation), or the minority/majority carrier

concentrations are altered so that the effect of surface recombination is minimized, e.g., via field-effect passivation. To understand the cause of this surface passivation, X-ray photoelectron spectroscopy (XPS) measurements were conducted. Samples treated with the phosphonic acids show suppression of $Pb^0$. Furthermore, we identify the formation of P-O-Pb bonds[31] through the emergence of a shoulder in the $Pb\ 4f$ signal at ~145 eV signal (Fig. 1c)[23,24]. All the PAs show a similar degree of $Pb^0$ suppression, albeit it is unclear whether the phosphonic acids protected the perovskite absorber from X-ray or vacuum-induced degradation during the measurements, which prevents the formation of $Pb^0$, or non-saturated $Pb^{2+}$ is chemically passivated with phosphonic acids which, in turn, reduces the amount of $Pb^0$. This raises two questions (1) Why did MPA not show any surface passivation effect, whereas BPA-derivatives, especially fluorinated ones, did? (2) Since $Pb^0$ suppression is common to all the tested PAs—Fig. 1c is $Pb^0$, in fact, an active defect center without the presence of $C_{60}$? The strength of the P-O-Pb signature is anti-correlated with the size of the molecule, where smaller molecules, being likely more mobile and causing less steric hindrance, appear to be creating denser layers. The highest P-O-Pb signal is observed for MPA, followed by BPA, FPBA, and pFBPA in decreasing order. At the same time, the size of the molecule is correlated with its effectiveness in passivating, as observed above, where smaller molecules passivate worse. To understand these phenomena, we turn to density function theory (DFT) calculations focusing on the surface of a halide perovskite in the presence of the PAs.

All reasonable phosphite orientations, including rotamers created by $CH_2R$-rotation, were considered to choose the lowest energy orientations. Stabilization of the perovskite surface is achieved due to the formation of the P-O-Pb covalent bond, as well as stronger hydrogen bonding of the phosphite with the formamidinium cations (P=O···HN) and weaker hydrogen bonding with iodine (P-OH···I). For R = $C_6H_5$ (BPA), $C_6H_4F$ (FBPA), and $C_6F_5$ (pFBPA), the phenyl ring adopts a horizontal position due to the weak halogen (C-F···I and intermolecular P=O···F-C), hydrogen (C-H···I and intermolecular P=O···H-C), and π···π bonding of phenyl with the formamidinium—Fig. 1d. The energy of substitution of IPb-I by IPb-$OPO_2CH_2R$ ($E_S$, eV) for each PA (12.5% of the surface iodide is substituted by the phosphite) can be seen in Fig. 1e, showing that surface substitution with phenyl-modified phosphites is energetically favorable, achieving $E_S$ = −0.41, −0.38, and −0.38 eV for R = $C_6H_5$, $C_6H_4F$, and $C_6F_5$, respectively, as compared to energetically unfavorable 0.09 eV for R = $CH_3$ in the case of 12.5% substitution. Surprisingly, despite the large size of the phenyl group, complete surface substitution is more energetically favorable, achieving $E_S$ = −0.41, −0.42, and −0.55 eV for R = $C_6H_5$, $C_6H_4F$, and $C_6F_5$ in the case of 100% substitution due to π···π bonding and the softness of the perovskite lattice allowing alternation of the $I_5PbO$ octahedra tiltings, see Fig. S1. Such differences in energy of substitution indicate superior surface passivation, thereby decreasing non-radiative losses for BPA-derivatives in contrast to MPA, Fig. 1e.

To further understand the interface passivation mechanism in the presence of the top-electrode stack ($C_{60}/SnO_x/Ag$) by the BPA molecules, we conducted Kelvin probe measurements to track changes in work function (WF). By measuring the WF under dark and light conditions, we extracted the surface photovoltage (SPV) in the presence and absence of $C_{60}$. With $C_{60}$, the SPV ($WF_{dark}$ - $WF_{light}$) follows the trend of the $V_{oc}$ for all BPA derivatives and control devices—Fig. 1f, g. HTL/perovskite/$C_{60}$ stacks were used in these experiments, and SPV values were measured at the electron-selective interface ($C_{60}$). The SPV amplitude increases by 155 mV with BPA, 215 mV with FBPA, and 225 mV with pFBPA. MPA leads to a decrease of 265 mV compared to the control device (Fig. 1f), suggesting that the electron selectivity at this interface is hampered, delivering the smallest surface polarization in DFT calculations, Fig. 1h. Increasing SPV amplitude in the presence of the full device stack (HTL and ETL) measured from the ETL suggests that the electron concentration in the $C_{60}$ layer is increased due to

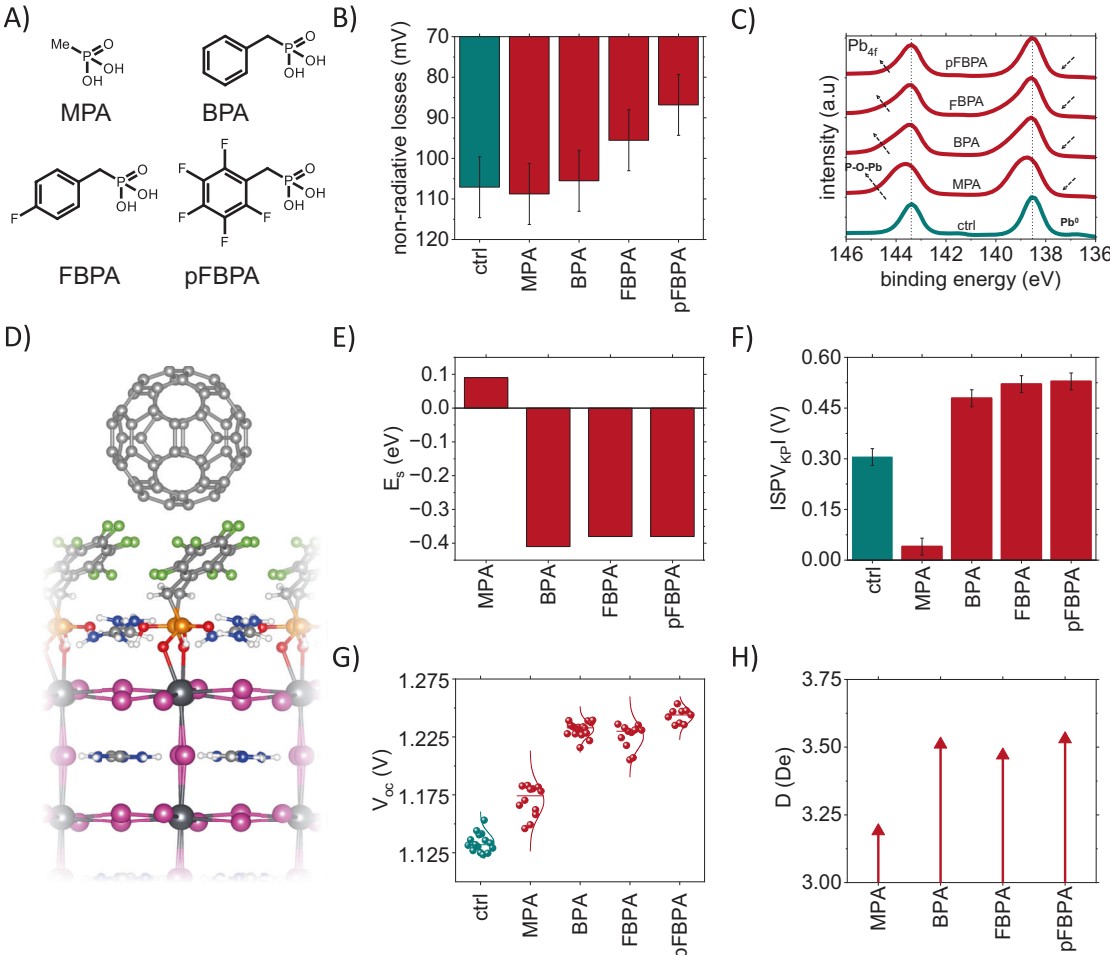

**Fig. 1 | Surface passivation of perovskite absorber with phosphonic acids.**
**A** Structural formulas of methylphosphonic acid (MPA), benzylphosphonic acid (BPA), fluorobenzylphosphonic acid (FBPA), and pentaflourobenzylphosphonic acid (pFBPA). **B** Non-radiative losses of half-cells with different surface treatments. **C** XPS Pb $4f$ spectra of layers with different surface treatments. **D** Side view of perovskite···$C_{60}$ interface with substitution of IPb-I by IPb-OPO$_2$HCH$_2$C$_6$F$_5$. **E** Energy of substitution of IPb-I by IPb-OPO$_2$HCH$_2$R ($E_s$, eV) for partial surface substitution (12.5%). **F** Surface photovoltage amplitude extracted from Kelvin probe measurements for ITO/HTL/perovskite/PA/perovskite samples. **G** Open-circuit voltage of single junction devices (1.65 eV) with different surface treatments. **H** Slab dipole (D, De) for the perovskite surface with partial surface substitution (12.5%) of MPA, BPA, FBPA, and pFBPA.

suppressed non-radiative losses at the perovskite/$C_{60}$ interface. The highest SPV amplitude (original signal is negative since p-i-n devices are measured from the n-side) is achieved with pFBPA, which benefits from simultaneous interface and surface passivation effects.

We then conducted UPS and KP measurements to compare control and pFBPA passivated devices. Extracting band energetics from UPS, the $E_{VBM}$ barely changed with pFBPA (from −5.7 eV to −5.65, within the margin of error)−Fig. S4. The $E_{VBM}$ offset increases slightly from 0.4 to 0.6 eV, and the WF stays constant around −4.9 eV. Since the difference between the $E_{VBM}$ and EF decreases with pFBPA, it signifies reduced n-type surface characteristics, which is not beneficial at the electron-selective interface. Despite an increasing $E_{VBM}$ offset, which should minimize the minority carrier concentration, decreasing recombination and reduced n-doping can also introduce a surface field, increasing minority carrier concentration. Hence, at this interface, due to these two compensating effects, we deem improved energetics causing the reduced recombination unlikely and used DFT to understand the interactions between perovskite, PAs, and $C_{60}$.

The densities of states of the interfaces of the phosphite-modified perovskite surfaces with $C_{60}$ do not show the emergence of recombination-dominating in-gap states originating from the phosphite (Fig. S2). This includes states stemming from the perovskite

being in contact with $C_{60}$ or any remote coupling, irrespective of the distance between the two. This is in contrast to previous results, irrespective of full or partial substitution at the perovskite; see Fig. S1. In the case of a partial pFBPA substitution at the surface, the charge transfer to $C_{60}$ appears to be still efficient, indicated by the mild negative Bader charge accumulation of −0.03 electrons (see Table S1) via the surface-aligned conductive π-system, likely preserving high FFs, despite introducing an additional spacer, Fig. S3. This indicates that a priori, the electrostatics at this interface align more favorably. Fortunately, the favorable alignment does not result in any additional resistive losses, unlike it is the case with dielectric layers[19]. In addition, the interface passivation also results in a higher slab dipole moment with systems with different PAs, Fig. 1h. The system with MPA exhibits the smallest slab dipole moment, 3.2 De. For the BPA derivatives, surface polarization increases, indicated by the slab dipole moment of ~3.5 De. Among them, the pFBPA system shows the highest slab dipole moment, albeit only marginally higher than the other BPAs. This enhanced surface polarization suggests the introduction of a more favorable dipole for electron extraction, which is also reflected as a change in the $E_{VBM}$ by 0.15 eV (Fig. S4).

After revealing the beneficial effect of the BPA-modified surfaces on the charge-accumulation efficiency, we aimed to understand if the

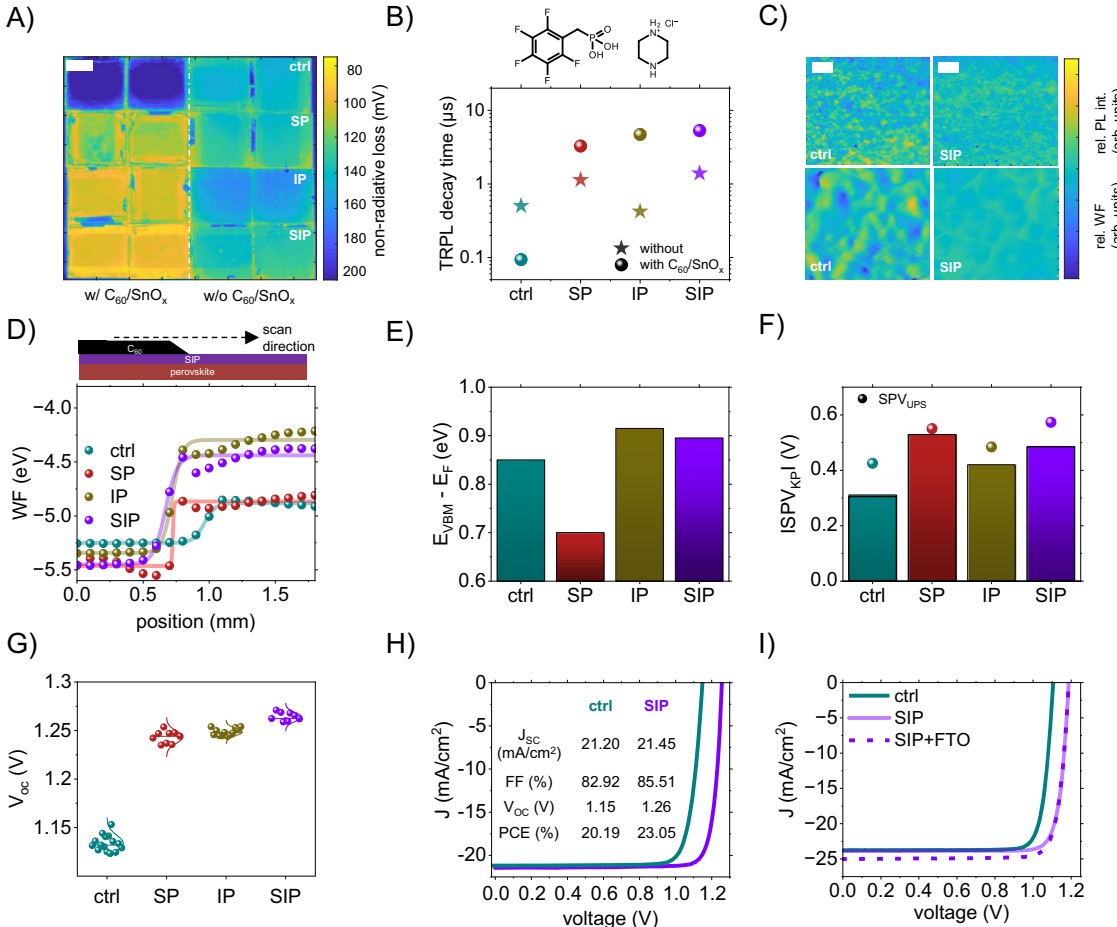

**Fig. 2 | Effective combination of surface and interface passivation. A** Non-radiative loss mapping measured using wide-field absolute PL (1 cm scale bar) with and without $C_{60}/SnO_x$ for different conditions, SP (pFBPA), IP (PCl), and SIP (PCl/pFBPA) on $2.5 \times 2.5$ cm² substrates. **B** Transient PL decay time ($t_3$) with and without $C_{60}/SnO_x$ for different conditions, SP, IP, and SIP. **C** Hyperspectral imaging (25 μm scale bar) and Kelvin probe force microscopy (250 nm scale bar) maps for control and SIP samples, where the z-axis represents the relative non-radiative loss and relative WF within the sample, respectively. **D** Work function results from scanning UPS measurements under light along graded perovskite/$C_{60}$ (control, SP, IP, SIP). **E** The distance between $E_{VBM}$ and $E_F$ for samples with control, SP, IP, and SIP. **F** SPV amplitude extracted from KP and UPS measurements for control, SP, IP, and SIP samples. **G** Open-circuit voltage of single junction devices (with 1.65 eV absorber) with different passivation, surface passivation (SP), interface passivation (IP), and surface and interface passivation (SIP). **H** Champion reverse scan JV curves of wide bandgap (1.65 eV) devices with control and SIP. **I** Champion reverse scan JV curves of medium bandgap (1.55 eV) devices with control and SIP on ITO and SIP on FTO-substrates.

effect can be translated into higher voltages in devices. To do so, we fabricated single-junction perovskite solar cells (ITO/Me-4PACz/SiOx-np/perovskite/passivation/$C_{60}/SnO_x$/Ag) and measured the photovoltaic device performance and PLQY to quantify the non-radiative recombination losses in full devices. On the one hand, all BPA derivatives (BPA, FBPA, pFBPA) achieve high $V_{oc}$ values above 1.225 V by passivating the $C_{60}$ interface and without compromising carrier transport, resulting in PCEs close to 22% in single junction devices (Fig. 1g and Fig. S3). The pFBPA devices also benefited from the initial increase in PL due to the surface passivation, achieving a $V_{oc}$ up to ~1.25 V and slightly outperforming other BPA derivatives (efficiency of 22%). On the other hand, MPA shows minor interface passivation, delivering a $V_{oc}$ up to 1.19 V, whereas the reference devices are limited to <1.14 V. The improvement (<40 mV) with MPA is attributed to the suppression of $Pb^0$ defects (Fig. 1c), which act as deep trap centers in the presence of $C_{60}$[32]. Even though promising performances (PCE ~22%, $V_{OC}$ >1.24 V) were demonstrated with pFBPA, these results suggest that there is still room for improvement by addressing the imperfect band energetics at the perovskite/$C_{60}$ interface, see Fig. S4. For that purpose, we aim to combine the surface passivation with a second interfacial modifier that allows for better interface passivation

(e.g., enhancing n-type behavior at the interface, introducing a favorable field effect, maximizing the $E_{VBM}$ offset, and minimizing the $E_{CBM}$ offset to improve the charge selectivity).

## Surface and interface passivation

To address across-interface recombination between the perovskite and $C_{60}$ by introducing a favorable dipole and n-type enhanced surface doping, we use a piperazinium halide molecule, similar to refs. 21,22, but with the counter ion, Cl⁻, which we have simulated to exhibit stronger binding to $C_{60}$ than the iodine salt (Fig. S1 and Table S1). Interestingly, piperazinium chloride (PCl), when utilized by itself, does not provide any surface passivation but rather results in a >10 mV increase in non-radiative losses without $C_{60}$ in place (Fig. 2a, right panel). When completing the stack, PCl strongly suppresses the interface recombination (Fig. 2a, left panel) when depositing $C_{60}$ and performs better than the best BPA derivative by itself (pFBPA); Figs. S5, S6. Hence, we propose to combine an interface passivation agent (IP) (e.g., PCl) and a surface passivation agent (SP) (e.g., pFBPA) with complementary passivation functionalities. When SP and IP are combined (defined as SIP), non-radiative recombination losses due to surface defects and interfacial recombination are strongly suppressed,

especially after $C_{60}/SnO_x$ deposition, see Fig. 2a. In contrast to the control devices with a PLQY of ca 0.1%, corresponding to an average non-radiative loss of 170 mV, the combined SIP passivation allows ~3% PLQY and thereby a low loss of ~90 mV in complete devices, Fig. 2a, less than the perovskite without the ETL, marking the combined SIP/ETL contact as a passivating contact.

We then measured the dynamics of charge carriers using time-resolved photoluminescence, as shown in Fig. S7. We select the illumination conditions to closely mimic the carrier densities under solar operation and set the laser fluence to generate a carrier density of ~$2 \times 10^{15}$ cm$^{-3}$. The measurements are performed on partial stacks with and without the ETL-stack ($C_{60}/SnO_x$) and illuminated from the ETL-interface to be more sensitive to this interface. In contrast to the control samples with ETL, which introduces an additional recombination channel, we observe an extension of the lifetime when the ETL-stack is deposited atop the SP/IP/SIP modified perovskite layers. The highest lifetime (5.3 µs) is achieved with SIP in the presence of the $C_{60}/SnO_x$, see Table S2. This concept is well-known from c-Si research and represents the scenario where a passivating contact improves minority carrier lifetimes[33]. We observe the same trend as in the PL mapping experiment, Fig. 2a. Furthermore, we analysed the data with the ETL-stack in line with ref. 34 and calculated the differential lifetimes, Fig. S8a, b. In the differential lifetime plots we observe two regimes, see Fig. S8c. In the initial phase, charge extraction is witnessed, where the fastest transfer occurs for samples with IP, followed by SIP. In the second phase, the plateauing indicates the onset of recombination, marked by a constant minority carrier lifetime, which is highest for SIP in line with the higher $V_{OC}$ in the devices. This means that IP facilitates fast charge transfer, slightly exceeding that of SIP. While the combined SIP has slightly slower extraction, recombination is further reduced, being the overall best condition for device performance, since the electron extraction is not limiting performance in these devices.

Moreover, PL mapping also revealed macro-scale homogeneity of the surface treatments across the samples. SIP samples exhibit homogeneous passivation, comparable to the former results, looking at a smaller area (~0.5 cm$^2$). The homogeneous passivation across the whole surface is highly encouraging for a scale-up of devices as it shows that the method appears to self-assemble into effective passivation layers, even for manufacturing-relevant dimensions of several cm$^2$ (Fig. 2a and Figs. S9–11). SP itself introduces inhomogeneity (standard deviation of NR-loss = 42 mV), especially in the presence of a top-electrode stack (Fig. 2a). However, when SP is combined with IP, i.e., SIP, samples show a more uniform distribution (standard deviation of non-radiative loss = 23 mV) and around an overall high QFLS (~1.29 eV) (Fig. 2a and Figs. S9, 10, 11) across the whole sample. We hypothesize that the excess PA that remains on the surface as an overlayer is washed away with the sequential application of IP.

After investigating the homogeneity on the cm scale, we aimed to understand whether the homogeneity is maintained at the micro-nano scale. When the WF distribution across the surface is investigated via Kelvin probe force microscopy (map size of $2 \times 2$ µm) and absolute PL spectra using hyperspectral imaging (map size of $100 \times 100$ µm), a decrease in the standard deviation of the WF (from 150 to 42 meV) and improved homogeneity in PL is observed (Fig. 2c). Improved homogeneity in the WF and PL distribution can contribute to minimizing scaling-up losses and achieving higher fill factors with minor optoelectronic fluctuations[20,35]. Furthermore, the changes in the surface microstructure are investigated with top-view scanning electron microscopy (SEM) and grazing-incident wide-angle x-ray scattering (GIWAXS), where it can be seen that with SP and SIP, the average domain size shows a slight increase (Fig. S12). We likewise observe a slightly improved crystallinity derived from the stronger signal of the "main" perovskite GIWAXS peak (as probed under every angle) in the case of SIP compared to the control films,

suggesting a slight improvement in the film's top morphology[20]. In contrast to many alternative surface modifications, we did not observe the formation of a low-dimensional (e.g., 2D) perovskite, or any other diffracting feature, suggesting the formation of a thin amorphous layer, Fig. S12.

## Surface energetics

To understand the influence of the surface treatments on the surface work function (under light and dark conditions, deducing surface photovoltage, SPV), we have conducted KP and UPS measurements for different layer stacks. We fabricated test samples ITO/HTL/perovskite/$C_{60}$ (graded) with specific geometry (Fig. 2d) and constructed an array (20 points) of measurements for UPS separated by 100 µm starting in the $C_{60}$ coated area and then ending in the perovskite zone while passing over the graded region (more details in the UPS-mapping section in Methods) to capture the band energetics starting from $C_{60}$ and down to the perovskite surface within a single sample. A similar sample geometry is utilized for KP measurements, but only two points on the $C_{60}$-coated area and non-$C_{60}$-coated are measured instead—Fig. S13. Exemplary output for the UPS spectra (secondary electron cut-off and $E_{VBM}$ and fits) and KP measurements can be seen in Figs. S14, S15, respectively. For all of the samples (control, SP, IP, SIP) in UPS measurements, the WF converges to similar values ~−4.9 eV in the dark and −5.4 ± 0.1 eV under light conditions—Fig. 2d) on $C_{60}$, which shows similar electronic characteristics of $C_{60}$ layers despite being deposited on different surfaces. When illuminated, the WF decreases with IP and SIP from −4.6 eV to −4.2 eV—Fig. 2d and Fig. S16. This does not necessarily signify a modulation towards more n-type without knowing how the band edges shift relative to $E_f$. Since the valence band maxima ($E_{VBM}$) also changes upon different passivation schemes—Fig. S16—we plot the distance between $E_f$ and the $E_{VBM}$—Fig. 2e—which shows increasing distance with the IP and SIP showing enhanced n-type behavior bringing the Fermi level closer to the conduction band. We observed similar direction and magnitude of shifts in KP results showing enhanced n-type behavior, which aligns with the literature[20]. At the electron extraction interface (perovskite/$C_{60}$), modulating the WF towards more n-type potentially decreases the minority carrier concentration (holes, $p_n$), resulting in lower non-radiative recombination. Hence, we conclude that the IP treatment does not alter the interface defect density but introduces a field effect that decreases $p_n$, which reduces recombination. For the SIP-treated samples, we observe the highest SPV amplitude (~0.57 eV for SIP and 0.41 eV for control), which is in line with the KP results, Fig. 2f. Also, the increase in the $SPV_{UPS}$ amplitude is similar to $SPV_{KP}$ (~0.16 eV), confirming higher carrier concentrations due to enhanced surface and interface passivation. Moreover, from DFT calculations (see Table S1), the Bader charge accumulation on $C_{60}$ increases further to −0.04 electrons with the partial (12.5%) PCl and pFBPA substitution, confirming the improved electron extraction efficiency.

The combination of pFBPA and PCl introduces a WF n-type shift, creating a favorable field effect at the electron extraction interface. This is complemented by the increased valence band maxima offset, minimizing the minority carrier concentration, Fig. S16. In addition to the changes in the surface work function, the valence band minima ($E_{VBM}$) is also measured across the perovskite/$C_{60}$ interface, as seen in Fig. S16. The $E_{VBM}$ offset increases significantly from (0.4 eV) with IP (1.05 eV) and SIP (0.95 eV) compared to the control devices, resulting in a higher offset for holes. A higher offset for minority carriers at the electron charge extraction interface is beneficial to minimize their concentration, decreasing non-radiative recombination. As a result, owing to the enhanced n-type surface characteristics and higher minority carrier offset of the IP, despite the SP treatment introducing a WF shift towards more p-type itself, the combination of SP & IP (SIP) preserves n-type modulation—Fig. S16.

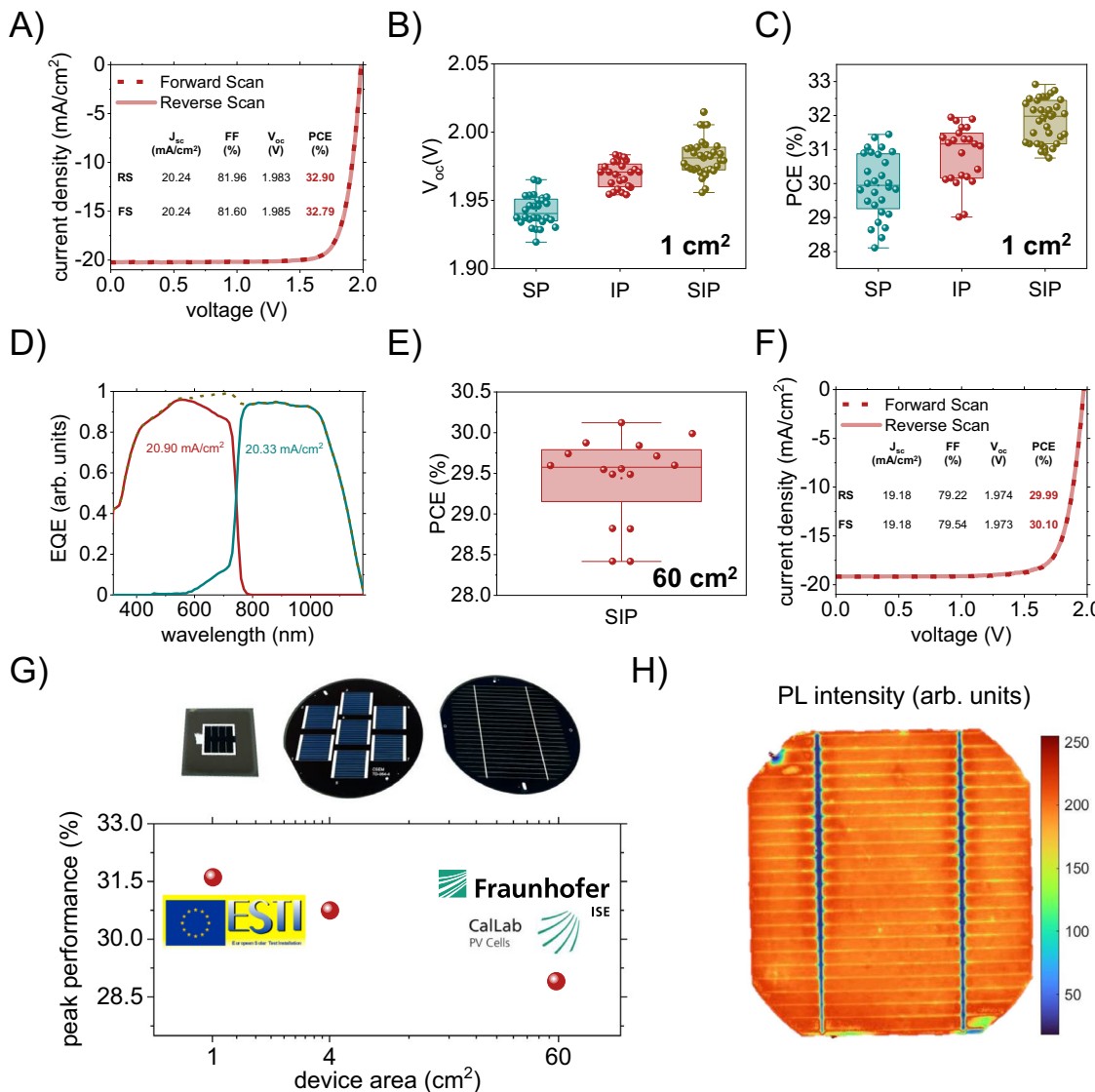

**Fig. 3 | Perovskite-Si tandem integration. A** Champion reverse and forward scan JV curve of the SIP device and PV performance parameters. **B**, **C** Open-circuit voltage and PCE (both reverse and forward scan) statistics of tandem devices fabricated with SP, IP, and SIP over multiple batches. **D** EQE spectrum of a tandem device with SIP (perovskite, Si, and the sum of each sub-cells). **E** PCE statistics of large area (60 cm²) devices with SIP. **F** Champion reverse and forward JV curves of a 60 cm² device and PV performance parameters. **G** Peak efficiency versus device active area graph and image of corresponding devices. 1 cm² device is certified by JRC-ESTI and 60 cm² device by Fraunhofer ISE CalLab. **H** PL imaging (top cell) of the 60 cm² tandem device showing the high level of top cell homogeneity, including the passivation.

## Single junction device performance

In single junction wide bandgap devices, the increase in open-circuit voltage with the utilization of SIP compared to SP or IP is ~25 and ~20 mV, respectively, Fig. 2g, with a champion PCE (SIP treatment) reaching 23%. In contrast, the control devices without any passivation yield ca. 20%– Fig. 2g, h. The significant improvement in pFF (up to ~88%, Fig. S17) originates from the decreased non-radiative losses, and FFs up to 85.5% (Fig. 2h), which is a result of increasing pFF and faster charge extraction at the ETL-interface, see Fig. S8c. We did not observe statistically relevant differences in the short-circuit current density (Fig. S18). To compare different combinations of PAs with PCl, we fabricated devices with BPA/PCl, which showed similar device performance to PCl alone. We explain this by noting that BPA shows no notable surface passivation compared to the other PAs (Fig. 1b), which does not enable improvements compared to PCl alone– Fig. S19. We further showcase the SIP strategy being applicable to other absorbers as well, by using a medium-bandgap absorber (1.55 eV) $Cs_{0.05}FA_{0.90}MA_{0.05}PbI_3$ (with an additional 3% $MAPbCl_3$)[12]

and demonstrate >24% PCE on ITO-substrates and >24.75% PCE on FTO-substrates, (Fig. 2i), (with only 0.22 mA/cm² discrepancy in $J_{sc}$ measured from EQE (24.79 mA/cm²) and JV (25.01 mA/cm²)– Fig. S20).

## Perovskite-silicon tandem device performance

We successfully integrated the newly developed SIP into the top cell of front-side polished perovskite-silicon-heterojunction tandem solar cells, Fig. S21a. First of all, on 1 cm², we fine-tuned the bandgap (~1.65 eV, with the final composition $Cs_{0.05}(FA_{0.9}MA_{0.1})_{0.95}Pb(I_{0.8}Br_{0.2})_3 + 3\%MAPbCl_3$), and increased the absorber thickness to 600 nm– Fig. S21a–for optimal performance in monolithic tandems. The improvement from the control devices to SIP was primarily driven by increased FF and $V_{oc}$. Our top-performing device reached a PCE of 32.9% (JV scan; with a $V_{OC}$ of 1.983 V, $J_{SC}$ of 20.24 mA/cm², and FF of 81.96%–Fig. 3a) and ~32.45% when tracking the MPP, Fig. S22. The certified steady-state efficiency of two similar devices is measured at 31.6 and 31.14% by JRC-ESTI, Figs. S23, 24. The open-circuit voltage achieved in the latter certification is 2.0006 V.

The passivation strategy is seamlessly transferred to the tandem devices, which results in a substantial increase in average PCE with SIP compared to SP and IP, respectively Fig. 3b, c. The EQE of one of the champion devices is depicted in Fig. 3d. The photogenerated currents for the perovskite top cell and the silicon-heterojunction bottom cell are 20.90 and 20.33 mA/cm², respectively, resulting in a cumulative photogenerated current of 41.23 mA/cm², excluding shadowing losses from the metal fingers (estimated as ~1.5%). Moreover, $V_{oc}$ and PCE distribution of the devices among six batches (including shunted devices) utilizing SIP can be seen in Fig. S25 and the best three batches (excluding shunted devices) in Fig. S26, which shows repeatable performance over hundreds of devices in different batches.

Furthermore, upon the successful integration of SIP into our 1 cm² tandem baseline and with the knowledge of homogeneous passivation, we then fabricated larger devices with 4 and 60 cm² active areas. Screen-printed metallization is utilized in devices with an area bigger than 1 cm² instead of evaporated Ag contacts, which require curing the paste at 130 °C for 10 min in ambient conditions[36]. The bottom cell processing was simplified compared to the 1 cm² bottom cells (no nc-SiOx(n) index matching layer and no SiOx-npback reflector). It is also worth mentioning that the certified 60 cm² champion device features a double-side sub-micron texture, Fig. S21b. The thickness of the absorber is increased to benefit from higher current mismatch due to the higher pFF of the bottom cell (towards the bottom cell, reducing the impact of possible shunts in the large area top cell)[24]. In order to withstand 130 °C curing in ambient for 10 min after screen-printing, PbX₂-excess amount is reduced from 6 to 3% causing 10–15 mV $V_{oc}$ reduction due to decreasing bulk absorber quality. The decreasing $V_{oc}$ due to PbX₂-excess reduction in the top cell is compensated by the suppressed perimeter recombination (since cell size becomes much larger than the diffusion length) in the bottom cell as the area increases from 1 to 60 cm², preserving $V_{oc}$'s ~1.97 V. The peak device performances achieved are 31.6% (certified by JRC-ESTI), 30.74% (in-house), and 28.90% (certified by Fraunhofer ISE CalLab) on 1, 4, and 60 cm², respectively (Fig. 3g). The in-house reverse and forward JV curves of the champion 60 and 4 cm² devices can be seen in Fig. 3f and Fig. S27, respectively. The maximum power point tracking of a 60 cm² device (Fig. S28) and the certified results (1.965 V, 18.87 mA/cm², 28.9% steady-state efficiency, 77.94% FF, 29.0% reverse scan efficiency and EQE) from Fraunhofer ISE CalLab can be seen in Fig. S29. The comparison of the EQE response of a 1 and 60 cm² device can be seen in Fig. S30, highlighting the superior optics of the small area cells due to a high-transparency top-electrode stack (e.g., thinner $C_{60}$, thinner and more transparent IZrO instead of ITO) despite thicker ALD-SnOx. Open-circuit voltages of ~1.96 V were achieved across these different length scales, showing that the surface and interface passivation is robust and homogeneous (from nanometer to cm scale) and transferable to larger areas, as confirmed by PL images of a large area device (Fig. 3h) with a tight intensity distribution (Fig. S31). Moreover, the average PCE from two batches composed of eight large-area devices is 29.5% (both reverse and forward scan data points are included since the hysteresis is not significant) (Fig. 3e), a testimony to the high repeatability of the SIP treatment.

## Stability—stabilization of the perovskite surface, but a weak interface, and enhanced ionic loss

Upon dark N₂ storage, a 1 cm² SIP device with an initial efficiency of 31.88% and two 60 cm² devices showed no sign of degradation (Fig. S32a, b), which aligns with our previous reports where similar hydrophobic fluorinated phosphonic acids were utilized[23,24]. While shelf stability is essential, solar cells must also withstand external stresses, particularly heat and light, to perform reliably under real-world operating conditions. Therefore, we focused next on testing the device under these operational conditions. To check the influence of the SIP

(control vs SIP) on the stability of the single junction devices, we tracked the maximum power point of devices in N₂, without encapsulation, at ~35 °C. After 115 h, the performance of the SIP devices decreased from 21 to 16, whereas the performance of the control devices increased from 17.5 to 20%, Fig. S33. Similar accelerated degradation trends are observed when either PCl or pFBPA is tested individually at the interface (see Fig. S34). We hypothesize that the performance loss of the SIP devices can be attributed to the ionic losses through the additional mobile ions at the interface (e.g., piperazinium⁺, Cl⁻, pFBPA⁻)[37,38], and a weakened perovskite/$C_{60}$ interface.

First, we conducted fast hysteresis (FH) JV and bias-assisted charge extraction (BACE) measurements to test the former hypothesis (see Figs. S35, 36)[37,38]. In the FH measurements, the significant losses in $J_{SC}$, FF, and consequently PCE observed in the SIP device between fast (ion-freeze) and slow (steady-state) measurements are attributed to enhanced ionic loss, see Fig. S35. In contrast, the control sample exhibited only minor performance losses between ion-freeze and steady-state conditions. The BACE measurements further confirmed the increased ionic losses in the SIP device, which shows an order of magnitude higher density of ions (Fig. S36). These results demonstrate that faster degradation of SIP devices (especially a strong decrease in $J_{MPP}$ over time, Fig. S38) is related to enhanced ionic losses, which are exacerbated under illumination in line with the observations from ref. 39.

Then, using DFT, we compared the binding energy ($E_b$, eV) of $C_{60}$ for different systems to further understand the accelerated degradation of devices with SIP, see Table S1. All reasonable binding sites were considered to choose the lowest energy site. We obtained ($E_b$) of −0.86 and −0.96 eV for partial (12.5%) and full (100%) surface substitution of IPb-I by IPb-OPO₂HCH₂C₆F₅ (pFBPA), respectively, indicating stabilization of the interface with denser surface coverage with the phosphites. The energetically favorable site for partial substitution is not above the phenyl rings but closer to the phosphite group, creating O···$C_{60}$ and F···$C_{60}$ contacts (in addition to the strong I···$C_{60}$ contact with the perovskite surface), see Fig. S1 and Table S1. This increase in negative $E_b$ amplitude signifies a stabilization of the surface and interface with pFBPA. Experimental results show that the interface stability changes significantly when piperazinium chloride is added onto the pristine perovskite surface. To understand this effect, we modeled the interface of a piperazinium chloride-modified perovskite surface with $C_{60}$. Substitution of FAI by HNC₄H₈NH₂·Cl (PCl) achieves $E_s$ of −0.07 eV, indicating stabilization of the piperazinium chloride-modified surface due to Pb-Cl bonds, stronger NH···Cl contacts created by HNC₄H₈NH₂ (as compared to NH···I contacts created by FA) and additional sp³ CH···Cl and CH···I contacts created by HNC₄H₈NH₂. However, the binding of $C_{60}$ to the perovskite surface is reduced ($E_b$ = −0.38 eV) indicating a destabilization of the piperazinium chloride-modified interface due to weaker N···$C_{60}$ contacts created by HNC₄H₈NH₂ (as compared to the O···$C_{60}$, F···$C_{60}$, and I···$C_{60}$ contacts created by the phosphite-modified surfaces) and inaccessibility of the Cl atoms of the perovskite because of capping by the piperazinium, see Fig. S1.

The presented DFT results imply that strong binding is created at halogen···$C_{60}$ contacts that could be realized either on top of a dense layer of halogenophenyl-functionalized phosphites (without piperazinium halide) on the fully substituted surface or near halogenophenyl-functionalized phosphites (with piperazinium halide) on the partially substituted surface, where they do not block halogen···$C_{60}$ contacts. The second case was modeled for a surface with partial substitution of Pb-I by IPb-OPO₂HCH₂C₆F₅ and one HNC₄H₈NH₂·Cl ion pair in the vicinity of an I···$C_{60}$ contact. Indeed, $E_b$ of $C_{60}$ reaches −0.99 eV, indicating a synergetic effect of the halogenophenyl-functionalized phosspite and piperazinium chloride as long as partial substitution is preserved, Fig. S1 and Table S1. Even

though experimentally determining the surface substitution fractions with the utilized concentrations (1 mM) is challenging, the significantly accelerated degradation of the devices might point towards high PCl fractions, lowering binding energies and causing a weaker interface.

To experimentally evaluate whether the destabilization at the interface due to the high fraction of PCl substitution is the main driver of accelerated degradation, we conducted a comparison of device stability by forming a SIP layer on top and below the perovskite layer (SIP-top and SIP-bottom configurations), as shown in Fig. S38a. Given that the SAM/perovskite interface is already of high quality (with a voltage loss of less than 15 mV), the SIP-bottom devices did not exhibit an enhanced open-circuit voltage. The accelerated degradation observed in SIP-top devices, where the perovskite/SIP/$C_{60}$ interface is present, was not seen in SIP-bottom devices, even though in this configuration, high-energy photons are absorbed closer to the SAM/SIP/perovskite interface, potentially providing more energy to drive chemical interactions. This difference in stability suggests that rather than the chemical reactions between the perovskite and SIP, the weakened perovskite/$C_{60}$ interface and enhanced ionic losses (also pointing towards the importance of $C_{60}$ in the ionic loss mechanism) are the primary cause of the accelerated degradation[40,41], making the devices more vulnerable to external stress factors such as light, humidity, and heat. We confirm this by peeling off, cleaning the top surface, and replacing the top-electrode stack (SIP/$C_{60}$/SnO$_x$/Ag), in an attempt to repair degraded SIP devices. Indeed, this approach partially restored performance, as shown in Fig. S38b. The PCE could be restored to > 80% of its initial value, indicating that full recovery is not possible, even when SIP and $C_{60}$ were redeposited, likely due to bulk degradation of the absorber, potentially made more vulnerable due to the weakened perovskite/$C_{60}$ interface and enhanced ionic losses. Reducing the concentration, and thus the surface fraction, of piperazinium chloride to preserve the halogen···$C_{60}$ contact that promotes high binding energy could be a potential strategy to achieve a better balance between efficiency and stability.

## Discussion

We believe that many passivation agents are, in fact, destabilizing the interface and, despite short-term improvements, are finally detrimental. It is noteworthy that recently ref. 25 in a similar device architecture, adding a chlorinated functional group to the benzyl ring of a passivating agent increased the binding energy at the interface with $C_{60}$. Moreover, ref. 42 further enhanced the stability of the interface by utilizing a chemically resilient head group (amidinium) instead of the commonly used ammonium, which retarded deprotonation reactions. Scheler et al.[39] modified the counter ion of the piperazinium halides to tosylate, and the piperazinium tosylate-modified devices suffered less from ion-induced losses. These points towards potential solutions to engineer the surface and interface passivating molecules towards stable and efficient perovskite photovoltaics, where the strong halide-$C_{60}$ interaction could be attached to a chemically passivating moiety, chemically resilient head groups can be utilized instead of ammonium groups, and ionic losses can be minimized with counter ion engineering.

## Methods
### Perovskite fabrication and deposition

1. **Triple-cation triple halide absorber (TC-TH) - wide (1.65 eV) bandgap perovskite absorber**

    Triple-cation perovskite (TC-TH) absorber utilized in this work is $Cs_{0.05}(FA_{0.90}MA_{0.10})_{0.95}Pb(I_{0.80}Br_{0.20})_3$ + ~3% $MAPbCl_3$. The absorber is fabricated with a one-step anti-solvent method. Three separate stock solutions are prepared, which are composed of 1.36 M FAI. (Dyenamo, >99.99%), 1.5 M PbI$_2$ (TCI, >99.99%) in 4:1 (volume ratio) DMF:DMSO and 0.68 M MABr

(Dyenamo, >99.99%), 0.68 M FABr (Dyenamo, >99.99%), 1.5 M PbBr$_2$ (TCI, >99.99%) in 4:1 (volume ratio) DMF:DMSO and 1.5 M CsI (Alfa Aesar, >99.9%) in DMSO. Then these solutions are mixed in a vial containing 15 mg/mL PbCl$_2$ (TCI, >99.99%) and 5 mg/mL MACl (Dyenamo, >99.99%) with an 800:200:40 volume ratio, respectively. The precursor solution (100 μL) is spread onto the substrate (2.5 × 2.5 cm²), then spun at 3500 rpm with 2000 rpm/s for 35 s. When 10 s are left during the spin-program, 250 μL of Anisole (Sigma-Aldrich, Anhydrous >99.8%) is dropped onto the substrate. Then the samples are annealed at 100 °C for 15–20 min inside the glovebox.

2. **Triple-cation medium (1.55 eV) bandgap perovskite absorber**

The medium bandgap (MBG) absorber utilized in this work is $Cs_{0.05}FA_{0.9}MA_{0.05}PbI_3$ with 5% excess PbI$_2$ and 5% excess MAPbCl$_3$. The solution is prepared by mixing, 1.68 M PbI$_2$, 0.048 M PbCl$_2$, 0.08 M MAI, 0.2 M MACl, 1.43 M FAI, 0.08 M CsI in DMF:DMSO 4:1 (volume ratio). The absorber is fabricated with the one-step anti-solvent method. The precursor solution (100 μL) is spread onto the substrate (2.5 × 2.5 cm²), then spun at 3500 rpm with 1000 rpm/s for 35 s. When 10 s are left during the spin-program, 250 μL of Anisole is dropped onto the substrate. Then the samples are annealed at 100 °C for 15–20 min inside the glovebox.

### Surface and interface passivation

For the PA molecule family study, 1 mM MPA (Sigma-Aldrich, >98%) or BPA (Sigma-Aldrich, >97%) or FBPA (Sigma-Aldrich, >97%) or pFBPA (Sigma-Aldrich, >97%) is dissolved in IPA and spin-coated at 6000 rpm dynamically and spun for 30 s followed by an annealing for 5 min at 100 °C.

For interface passivation, 1 mM PCl (Dyenamo, >98%) in IPA, for surface passivation, 1 mM pFBPA (Sigma-Aldrich, >97%) in IPA and for the surface and interface passivation, 1 mM PCl and 1 mM pFBPA in IPA were utilized. Upon deposition of the perovskite absorber, the SP, IP, or SIP is spin-coated at 6000 rpm dynamically and spun for 30 s, followed by an annealing at 100 °C for 5 min.

For the passivation of medium bandgap perovskite, piperazinium iodide (Dyenamo, 1 mM in IPA) or propane-diammonium iodide (TCI, 2 mM in IPA) is utilized instead of piperazinium chloride.

There are two other methods to apply SIP and reach similar performance.

1) To use 1 mM pFBPA as an additive in the perovskite ink that accumulates on interfaces[24] (demonstrated in our previous work), providing surface passivation and then 1 mM PCl is spin-coated on top of the absorber with pFBPA additive.

2) To use sequential application of 1 mM pFBPA followed by 1 mM of PCl on the top surface. Samples are annealed in between PCl and pFBPA at 100 °C for 5 min. Likewise, a 6000 rpm spin recipe is utilized.

### Solar cell fabrication

**Fabrication of single-junction perovskite solar cells.** A 0.7 mm ITO (15 ohm/square - TKing) coated glass substrates are cleaned with Helmanexx and DI, respectively, using ultrasonic baths for 15 min. Samples are exposed to UV-ozone for 30 min right before the SAM deposition. SAM (e.g., Me-4PACz) solution (1 mg/mL) is spin-coated onto substrates for 100 μL with a program of 10 s resting time, 3000 rpm, 300 rpm/s. Then the substrates are annealed for 10 min at 115 °C. To improve the wetting, 0.1% SiOx-np in EtOH (20 nm diameter) spin-coated for 100 μL with a program of 2000 rpm with 500 rpm/s for 30 s and annealed for 115 °C for 10 min[24]. For the deposition of the perovskite absorber, the one-step absorber (TC-TH) that is mentioned in the above sections is utilized. Details of the passivation solutions (IP, SP, or SIP) have been given in the section above. About 20–25 nm of $C_{60}$ (Creaphys, >99.9%−resublimed) is deposited via thermal

evaporation with a rate of 0.2 A/s in the Angstrom thermal evaporator. Then ALD-SnO$_x$ is deposited at 95–100 °C with pulse purge times of 0.3 s/6 s/0.1 s/6 s for 175 cycles with TDMASn and H$_2$O sources. The utilized thicknesses for baseline devices are 20–22.5 nm, measured with an ellipsometer on c-Si. For the metal contact, 150 nm of Ag is deposited via thermal evaporation with a rate of 1.5-2 A/s. 110 nm of MgF$_x$ was evaporated on the glass side for anti-reflective coating. The device active area is 0.2 cm$^2$.

### Fabrication of monolithic perovskite-silicon-heterojunction tandem solar cells – 1 cm$^2$

The bottom cell fabrication process is performed on 200 or 280-um-thick n-type, 2 ohm.cm, FZ Si wafers. A layer of SiNx is applied to the front side using PECVD, and the rear side is textured using a KOH solution to create a random pyramid pattern. The SiNx masking layer is then removed, and samples undergo a cleaning sequence prior PECVD process. Through PECVD, hydrogenated amorphous and nanocrystalline Si layers are deposited on both sides. On the front side, a-Si:H(i), a-Si:H(n), nc-Si:H(n), and nc-SiOx:H(n) are deposited at 200 °C. On the rear side, a-Si:H(i) is deposited at 200 °C, followed by ultra-thin SiOx and nc-Si:H(p) layers at 175 °C[43]. On the rear side, a 40-nm-thick ITO is deposited by sputtering through a shadow mask. The samples then receive a SiO$_2$-np coating on the pyramid textures acting as an infrared rear reflector, which is then capped with an Ag layer through the same shadow mask used for the ITO deposition[44]. A 500-nm-thick SiOx layer covers the rear side, including both metalized and non-metalized regions. Additional Ag is sputtered onto the active area to finalize the rear contact. The front side is completed by sputtering a 20-nm-thick ITO layer for interconnection. As the final step before top cell processing, 4" wafers are then laser-cut into 2.5 × 2.5 cm$^2$ substrates, which are annealed at 210 °C for 30 min in ambient to optimize the properties of the solar cells.

Me-4PACz (TCI or Dyenamo) is spin-coated with the same spin recipe utilized in single-junction devices. In tandems (for 1, 4, 60 cm$^2$), 30 μL/mL DMF is added into the SAM solution to improve solubility and avoid micelle formation. After SAM deposition, the surface is washed by spinning 150 μL ethanol at 3000 rpm to clean residual SAM and DMF, followed by annealing at 115 degrees. To improve the wetting of the perovskite precursor on Me-4PACz, 0.1% wt SiOx-np (~20 nm diameter) in ethanol or 0.1% wt AlOx-np (Sigma-Aldrich, <50 nm, 20 wt% in isopropanol) diluted (1:200) solution in isopropanol is spin-coated statically at 2000 rpm with 500 rpm/s acceleration, followed by an annealing step of 10 min at 115 °C. The absorber utilized in tandems is Cs$_{0.05}$(FA$_{0.9}$MA$_{0.1}$)$_{0.95}$Pb(I$_{0.80}$Br$_{0.20}$)$_3$ + 3% (MAPbCl$_3$). The absorber is fabricated with the one-step anti-solvent method. The molarity of the solutions is increased to 1.7 M from 1.5 M to adjust the thickness of the absorber by keeping the PbI$_2$ and PbBr$_2$-excess amount constant. When 10 s are left during the spin-program, 250 μL of Anisole (Sigma-Aldrich, >99%) is dropped onto the substrate. Then the samples are annealed at 100 °C for 15–20 min inside the glovebox. The thickness of C$_{60}$ is reduced to 15–20 nm for the tandem devices, and 175 cycles of ALD-SnO$_x$ is used in tandems. After the ALD-SnO$_x$, 35 nm of IZrO with sheet resistance around 225 ohm/square is deposited from a 4-inch 98% I$_2$O$_3$ + 2% ZrO target with deposition power of 70 W and working pressure 2.7 μbar with ~0.14% of O$_2$ to Ar ratio. After the TCO, ~500 nm of Ag grid is evaporated with a rate of 2 A/s, which is followed by 100 nm evaporation of LiF or MgF$_x$ with a 1 A/s rate in an Angstrom thermal evaporator.

### Fabrication of monolithic perovskite-silicon-heterojunction tandem solar cells larger than 1 cm$^2$

The 60 cm$^2$ devices employed double-sided sub-micron textured bottom cells. On the front side, a-Si:H(i) and a-Si:H(n) were deposited, while a-Si:H(i) and a-Si:H(p) layers were added on the rear. Solution-processed layers (SAM, SiOx-np, perovskite, SIP) are spin-coated on a

4-inch bottom cell. After 30 min UVO treatment, samples are transferred into the glovebox. A 1:1 mixture of Me-4PACz:DBC(4PADCB, Dyenamo) was spin-coated statically on the 4-inch wafer. After 115–120 °C annealing for 10 minutes, EtOH was spin-coated to wash the SAM excess. Then, a diluted suspension of SiOx-np was spin-coated and annealed at 115–120 °C for 10 minutes to form a partial layer to improve the wetting of the perovskite ink. About 1 mL of perovskite ink was spread across the 4-inch wafer and spin-coated (at 1000 rpm for 15 s and 4000 rpm for 30 s) before dripping anisole to initiate the crystallization. The samples were then annealed at 105 °C for 20 min. pFBPA (1 mM) was added as an additive to the perovskite ink, while PCl was spin-coated and annealed at 100 °C for 5 min to complete the SIP treatment

About 25 nm of C$_{60}$ was evaporated, and 125 cycles of ALD-SnO$_x$ was deposited using an Oxford Instruments ALD tool. 45 nm ITO with R$_{sh}$ 130 ohm/sqr was sputtered on top of SnOx. Instead of the evaporated metal contacts, screen-printed Ag (~20 μm) cured at 130 °C was used[36]. To withstand curing, the PbI$_2$ excess in the ink was decreased from 6% to 3%. Indeed, triple-cation triple halide (CsFAMA) absorbers have shown enhanced resilience to thermal stress compared to CsFA-triple halide absorbers (MACl addition is not considered as MA-inclusion in this work) (Fig. S39). The shadowing losses from the metal contacts for the 60 cm$^2$ metallization design are estimated at ~2.85%, which is higher than the 1 cm$^2$ design (~1.5%).

### Characterization

**Density functional theory.** Quantum Espresso was used for modeling by plane wave density functional theory[45,46]. The wavefunctions were constructed using norm-conserving scalar-relativistic pseudopotentials with Pb 5$d^{10}$6$s^2$6$p^2$ and I 5$s^2$5$p^5$ valence electrons[47]. The kinetic energy cutoff was set to 80 Ry and the charge density cutoff to 320 Ry. The Brillouin zone was integrated on a Monkhorst-Pack k-grid with 50 k-points per Å$^{-1}$, and the revised Vydrov–Voorhis (rVV10) electron density functional was adopted to properly account for the non-covalent bonding[46–50]. In the structure relaxations, the atomic positions were updated using the Broyden–Fletcher–Goldfarb–Shanno algorithm until the residual forces were reduced to ≤10$^{-4}$ Ry Bohr$^{-1}$.

(0 0 1)-oriented FAPbI$_3$ slabs with FAI-termination and eight surface formula units were modeled, see Fig. S2. A vacuum layer of 25 Å thickness was employed with a saw-tooth electrostatic potential centered in the middle of the vacuum layer for dipole correction.

The energy of substitution of IPb-I by IPb-OPO$_2$HCH$_2$R was calculated as E$_s$ = (E$_{perovskite-OPO2HCH2R}$ + xE$_{PbI2}$ − E$_{perovskite-I}$ − xE$_{IPbOPO2HCH2R}$)/x, where x = 1 and 8 for 12.5 and 100% substitution, respectively, and IPb-I and IPb-OPO$_2$HCH$_2$R are molecules, according to the reaction scheme perovskite-I + IPb-OPO$_2$HCH$_2$R → perovskite-OPO$_2$HCH$_2$R + PbI$_2$. This descriptor was chosen due to the fact that phosphonic acids are strong enough to dissociate in the polar solvent, providing phosphite anions to create Pb(II) phosphites that modify the perovskite surface. The energy of substitution of FAI by HNC$_4$H$_8$NH$_2$·Cl was calculated as E$_s$ = (E$_{perovskite-Cl·H2NC4H8NH}$ + 8E$_{FAI}$ − E$_{perovskite-I·FA}$ − 8E$_{HNC4H8NH2·Cl}$)/8, where FAI and HNC$_4$H$_8$NH$_2$·Cl are tight ion pairs, according to the reaction scheme perovskite-I·FA + HNC$_4$H$_8$NH$_2$·Cl → perovskite-Cl·H$_2$NC$_4$H$_8$NH + FAI.

The binding energy of C$_{60}$ with the perovskite surface was calculated as E$_b$ = E$_{perovskite···C60}$ → E$_{perovskite}$ − E$_{C60}$, where C$_{60}$ is a molecule, according to the reaction scheme perovskite + C$_{60}$ → perovskite···C$_{60}$.

**JV.** The 4PP measurement method is used. JV measurements were obtained using a two-lamp (Halogen and Xenon) class AAA WACOM sun simulator with an AM1.5 G irradiance spectrum at 1.000 W/m$^2$. 0.1 cm$^2$ shadow masks are used to measure cells with an area of 0.2 cm$^2$ unless otherwise stated. Opaque devices are illuminated from the glass side. The single junction cells are measured with a scan rate of ~0.50 V/

s, first, a reverse scan and then a forward scan. Both reverse and forward scans are included in the JV statistic plots. Three-point weight MPP measurements are performed using an in-house written LabVIEW code. For the measurement of tandem devices, before each measurement, the calibration of the AAA WACOM system is checked with three different certified cells with different spectral responses to minimize the spectral mismatch of the sources. Initially, to adjust the intensity of the Xe lamp, a calibrated encapsulated cell (WPVS−blue cell) containing a monocrystalline silicon solar cell, which is covered with a KG5-filter was used. Then, to adjust the intensity of the Ha lamp, a calibrated encapsulated cell (WPVS−red cell) containing a monocrystalline silicon solar cell, which is covered with an RG780 filter, was used. Then, the overall spectrum is checked with a calibrated encapsulated cell (WPVS without filter broadband−black cell). To further fine-adjust the intensity of the Ha lamp, a tandem cell within the fabricated batch was first measured by EQE, then the lamp intensity was adjusted to match the integrated EQE of the bottom cell with the measured Jsc, notably with the conditions in which the bottom cell is the current-limiting cell. All calibrated Si cells are provided by Fraunhofer ISE. Then the cells with ~1 cm² are measured with a scan rate of ~0.1 V/s, and a similar MPP tracking algorithm is utilized as the single junction perovskite solar cells. A temperature-controlled (25 °C) brass chuck was used.

All IV measurements for devices bigger than 1 cm² were made on a large area class AAA+ solar simulator from WACOM using two light sources (xenon and halogen). The spectral response and overall light intensity of the simulator were verified with externally calibrated reference cells to approximate 1-sun AM1.5 G equivalent intensity. Initially, to adjust the intensity of the Xe lamp, a calibrated encapsulated cell (WPVS−blue cell) containing a monocrystalline silicon solar cell, which is covered with a KG5-filter was used. Then, to adjust the intensity of the Ha lamp, a calibrated encapsulated cell (WPVS−red cell) containing a monocrystalline silicon solar cell, which is covered with an RG780 filter, was used. Then, the overall spectrum is checked with a calibrated encapsulated cell (WPVS without filter−black cell). All calibrated Si cells are provided by Fraunhofer ISE. All measurements were done using a four-point method. All cells were measured on a metallic vacuum chuck with an active temperature-controlled set to 25 °C. Cell areas were masked with laser-cut aperture masks whose physical areas were optically measured. All IV curves were scanned between 2 and 0.1 V. At a scan rate of ~80 mV/s. Maximum power point tracking was achieved using a three-point algorithm, which actively modified the voltage throughout the measurement. Small areas (4 cm²) were contacted using two-Kelvin probes on either side of the metal grid. Large area cells (60 cm²) were contacted using 4-Kelvin probes with one contacting each end of each bus bar. The contact point was ~2 cm from the end of the bus bar. Shading due to probe bars is corrected after the measurements by application of the correction factor (x1.016) to Jsc.

**Long-term stability.** Cicci Research stability measurement setup is utilized where the samples are kept in N₂ atmosphere, under 1-sun illumination by tracking the MPP with the perturb & observe algorithm. The N₂ chambers are temperature-controlled, and cells saturate ~10–15 °C higher than the chamber temperature.

For stability testing, the PbX₂ excess amount is reduced to 3% from 6% and 2PACz HTL is utilized for simplicity.

**EQE.** EQE spectra were measured with a custom-made spectral response set-up where the samples were irradiated with chopped light at a frequency of ~200 Hz and the response was measured with a lock-in amplifier. For tandem cells, visible and IR light biases are used to saturate complementary sub-cells and to measure each sub-cell near short-circuit conditions. 0.7 and 1.2 V bias voltages are applied to the cell when measuring top and bottom cells, respectively.

**FH Measurements.** JV curves were obtained by applying a triangular voltage pulse to the cells, starting approximately at the V$_{OC}$. This was followed by a reverse sweep from open-circuit to −0.1 V and a forward sweep from −0.1 V back to V$_{OC}$ at variable frequencies or scan speeds (V/s) using a system developed by FastChar UG[37,38].

**BACE.** The device was initially biased at a voltage equivalent to the open-circuit voltage. After a delay time of 30 s, a bias of 0 V was applied to extract the injected and capacitive charge within the device. The transient current was recorded using a Keithley 2400 instrument controlled by a custom LabView program. The extracted charge was calculated by integrating the transient current, while the charge carrier density was determined by dividing the total charge by the elementary charge and the cell volume[37,38].

**SEM.** SEM images were acquired with acceleration voltages ranging from 1 to 5 kV using either an Everhart-Thornley or an in-lens detector (JEOL JSM-7500TFE or Zeiss NVision 40 microscopes).

**Steady-state PL.** A custom-built steady-state PL setup is utilized to measure in situ PL and PLQY. The light from the laser diode is coupled into a fiber, directed into the integrating sphere and the sample is illuminated. Then the emission from the sample is homogenized by multiple reflections within the integrating sphere and coupled out into another fiber that is connected to a spectrometer. As long as the emission and the excitation, which in our case the values are (750 to 800 nm) and (532 nm), careful comparison of the peaks enables us to measure PLQY (down to 10⁻⁶). The OD filter wheel helps to change the light intensity between 0.01 sun and 3 sun. As a calibration check, three fluorescent test samples with high specified PLQY (~70%) supplied from Hamamatsu Photonics were measured, where the specified value could be accurately reproduced within a small relative error of less than 5%. For QFLS measurements, the light intensity is tuned to 1-sun depending on the bandgap of the absorber using an external c-Si photodetector.

**Intensity-dependent steady-state PL.** The samples were illuminated in the steady-state PL setup as described above. A neutral density filter wheel was used to attenuate the laser power to measure at different intensities which was compared to the initial measured. The samples were illuminated at a given intensity for 3-s illumination time using an electrical shutter.

The pseudo-J-Vs were deduced from the intensity-dependent QFLS measurements. This was done by calculating the dark-current density from the generated current density at a given light intensity in equivalent suns. For example, 1 sun corresponds to 21.0 mA/cm², and 1% of one sun to 0.210 mA/cm². The obtained dark current was then plotted against the measured QFLS at the given light intensity to create a transport-free dark J-V-curve, which was then shifted to the J$_{sc}$ in the JV-measurement to create the pseudo-JV curve, allowing for reading of the pseudo-FF and V$_{oc}$ of the measured partial cell stack (e.g., glass/absorber, or half-cell, or the complete cell).

**Hyperspectral imaging (HI).** Two hyperspectral imaging (HI) systems with different fields of view were used. Both record luminescence intensity signal along three dimensions {x,y,λ}.

The wide-field (~10 cm scale, 130 μm spatial resolution) hyperspectral photoluminescence images were acquired with Grand-EOS equipment from PhotonEtc (2 nm spectral resolution). A wide-field illuminating LED system at 525 nm homogeneously brightens the samples at a power of 42 mW/cm² (~0.6 Sun for 1.65 eV bandgap).

Micro-scale (~1 μm spatial resolution) hyperspectral photoluminescence images were acquired with a home-built microscope with Thorlabs optomechanical elements, a 2D bandpass filtering

system from company PhotonEtc with 2 nm resolution, and a 4Mpix silicon-based sCMOS camera Hamamatsu ORCA Flash v4. The sample is illuminated with a 405 nm LED at varying powers from 75–750 mW/cm² (~1–10 Sun for 1.65 eV bandgap) through an infinity-corrected ×50 Nikon objective with numerical aperture of 0.6. The luminescence is collected through the same objective. The excitation beam and luminescence signals are separated with an appropriate dichroic beamsplitter and filters.

Both imaging systems are absolutely calibrated, and the obtained absolute PL spectra are fitted according to Generalized Planck's law[51] to obtain the quasi-fermi level splitting (QFLS) and bandgap maps. The QFLS are corrected to match 1 sun fluence condition, based on illumination intensity, bandgap of the samples and optical ideality factor extracted from illumination power dependent broadband PL images.

Measurements were performed under a nitrogen atmosphere at room temperature.

**Bichromatic PL mapping.** Photoluminescence imaging was performed using a custom setup. Excitation of the perovskite layer is done using an array of blue light-emitting diodes (450 nm central wavelength), and effective generation is roughly equivalent to 1 sun. The sample emission is filtered using long-pass filters and imaged using a Thorlabs Kiralux CCD camera mounted with a 6 mm focal length objective. Reference of the filters in case: Thin film filter for PK PL: ThorLabs FELH-0650 and Absorbing filter for PK PL: SCHOTT RG-695 and SCHOTT RG-715.

**Transient photoluminescence.** The samples are illuminated by an EKSPLA ATLANTIC25-8UV pulsed laser at 532 nm emission wavelength and 50 kHz repetition rate, the laser beam being shaped with a rotating diffuser and a Kohler-like optical lenses for homogeneous illumination (86 uW in average). The homemade microscope with Thorlabs optomechanics is equipped with a 650 nm cut-off wavelength beamsplitter (Thorlabs DLMP650R) allowing illumination and collection, an objective X10 Olympus NA = 0.25, and Thorlabs FELH550 and FGL665 longpass filters in the collection branch. The illumination spot size is 3.5 mm. The luminescence is recorded by an intensified electron-multiplied CCD camera (em-ICCD, PIMAX4, Princeton Instruments). The gate width of the camera is set to 3 ns, and the decays are recorded every 15 ns from 0.5 μs before to 14.5 μs after the laser pulse. The decays are obtained by spatially averaging the experimental data (x, y, time) over the sensor size.

**Kelvin probe (KP).** KP measurements were performed using a KP Technology Ltd. SKP5050 probe using a 2 mm gold tip. The work function of the samples was measured during three cycles of dark and illumination (10 min each) on at least two different points. The illumination was performed with a white-LED previously calibrated with a silicon diode to obtain 1 sun of intensity. The SPV of the samples was calculated by the difference in the average work function under dark and illumination. Illumination source is white-LED with an intensity of ca. 1 sun calibrated with a Si photodiode.

**Kelvin probe force microscopy (KPFM).** KPFM maps were generated using an AFM Dimension Edge Bruker microscope with a Pt-coated Si tip (Model SCM-PIT-V2), having a spring constant $(k) = 3\,N\,m^{-1}$. Scans were performed over 2 μm at 512 pixels and 1 Hz frequency in a dual-pass setup, the first pass to record topography and the s to measure the contact-potential difference (CPD) with a tip potential of 5 V (both passes are in tapping mode). The WF of the Pt tip was precisely calibrated by measuring the CPD relative to a freshly cleaved highly oriented pyrolytic graphite (HOPG); a material with a standard work function of about 4.6 eV. This allowed us to estimate locally the surface potential values of the different perovskite films with high spatial resolution.

**XPS/UPS and UPS-mapping.** XPS measurements were performed using an Axis Supra (Kratos Analytical) with a monochromatic Al Kα X-ray line. The pass energy was set to 20 eV with a step size of 0.1 eV for XPS, and to 40 eV with a step size of 0.15 eV for AR-XPS. The emission current was set to 15 mA. The samples were electrically grounded to limit charging effects. UPS was measured using a He I UV source, with the pass energy set at 10 eV and a step size of 0.025 eV. The intensity of the UV source, together with the measurement area, was selected in order to prevent the energy analyzer from saturating. Secondary electron cutoff (SECO) was used to assign the work function of the material, while the valence band onset was used to estimate the valence band maximum with respect to the Fermi level.

For the UPS-mapping experiment (Fig. 2d), to extend the evaporation edge formed while evaporating $C_{60}$, the evaporating mask is placed 5 mm away from the substrate instead of 1 mm (noted as graded $C_{60}$ in the main text).

**Grazing incidence wide-angle X-ray scattering (GIWAXS).** Data were acquired at a 2° incidence angle using a point-collimated beam on a Bruker D8 Discover Plus diffractometer equipped with a Copper rotating anode and an Eiger2 500 K detector. The beam was shaped by a focusing Göbel mirror, followed by a 0.3 mm pinhole, and then collimated by a 0.2 double-pinhole collimator. The detector was mounted at 117 mm sample-detector distance in gamma-optimized orientation, to collect one quadrant by taking three separate images, which were then merged into the shown images using Diffrac.EVA. Raw data images were converted to Q using GIXSGUI[52].

A number of repeating rectangular shapes are visible on the GIWAXS images, these shapes stem from Kapton tape used to cover minor damage on the detector, and they have no impact on data interpretation.

### Reporting summary

Further information on research design is available in the Nature Portfolio Reporting Summary linked to this article.

## Data availability

Source data are provided with this paper. All other data of this work are available from the corresponding authors on request.

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

## Acknowledgements

The authors thank Dr. Peiliang Chen from Scenergy and Patrick Wyss for wet chemical processing of the Si wafers, Joël Spitznagel, and Sylvain Dunand for the 1 cm² Si bottom cell fabrications, Julien Gay for the SiOx-np supply, Adrien Theytaz and Jean-David Decoppet for SnO$_x$ atomic layer deposition and screen-printing for the large area tandems, Antoine Descoeudres, Vanessa Gainche, and Bertrand Paviet-Salomon for the fabrication of 4 and 60 cm² Si bottom cells, Gabriel Christmann for PL

imaging of 60 cm$^2$ devices, Pascal Alexander Schouwink for GIWAXS measurements and analysis, and Jeong Kwon for supporting the development of large-area tandems. The authors acknowledge funding from the European Union's Horizon 2020 and innovation program (VIPERLAB, 101006715), the European Commission and the Swiss State Secretariat for Education Research and Innovation (SERI) (TRIUMPH - 101075725 and PEPPERONI - 101084251), the Swiss National Science Foundation (PAPET, 200021_197006; A3P, 40B2-0_1203626, Radicals, CRSII5_216647), the Swiss Federal Office of Energy (PRESTO, PERSISTARS, BESTOBOT), (COMET, 502791-01) and the ETH Domain through an AM grant (AMYS). M.O., D.T., and A.K. acknowledge funding from the European Union's Horizon 2020 research and innovation program under a Marie Skłodowska-Curie grants (945363 and 101034260). D.T. acknowledges the State Secretariat for Education, Research, and Innovation for an FCS/ESKAS Swiss Government Excellence Scholarship. F.L. and A.F.C.M. thank the BMBF for funding (03EE1183C). F.L. thanks the Volkswagen Foundation for funding via the Freigeist Program. The research reported in this publication was supported by funding from King Abdullah University of Science and Technology (KAUST). For computer time, this research used Shaheen III and Ibex, managed by the Supercomputing Core Laboratory at KAUST.

## Author contributions

Conceptualization of the idea by K.A. and F.S. Experimental design by K.A. and C.M.W. Single junction and tandem top cell fabrications (1 and 60 cm$^2$) by K.A. DFT calculations and analysis by A.O. and U.S. PLQY measurements and analysis by K.A. 1 cm$^2$ bottom cell development and fabrication by D.T. and J.H., and 60 cm$^2$ bottom cell development and fabrication by C.A. XPS/UPS measurements and analysis by M.M. and K.A. Large area tandem developments (>1 cm$^2$) by K.A., D.A.J., M.D.B., and L.C. KPFM and SEM by M.O., GIWAXS analysis by A.G.K. KP measurements by A.-F.C.-M. and F.L. Wide-field PL and HI measurements by A.L., J.B.P., and D.O. Writing, review & editing K.A. and C.M.W. Supervision and funding acquisition by A.W.H., Q.J., C.B., and C.M.W. Manuscript revision, all authors.

## Competing interests

The authors declare no competing interests.
