## [Transparent Peer Review file · Nature Communications]

60 cm² perovskite-silicon tandem solar cells with an efficiency of 28.9% by homogeneous passivation

Corresponding Author: Dr Christian Wolff

Version 0:

Reviewer comments:

Reviewer #1

(Remarks to the Author)

The authors propose a bimolecular passivation strategy to achieve both field-effect and chemical passivation at the C₆₀ interface. A thorough explanation of the underlying mechanism is provided. As a result, the perovskite/silicon tandem solar cells, particularly the 4 cm² and 60 cm² devices, reach impressive stabilized power conversion efficiencies of 30.74% and 28.9% (certified), respectively. This work is of clear significance to the field of perovskite/silicon tandems, especially with regard to large-area scalable fabrication. Most of the authors' conclusions are well supported by the data, including DFT calculations, SPV measurements, PL mapping, and photovoltaic device performance. The methodology is generally sound and well described. However, certain aspects—particularly related to large-area device fabrication—would benefit from additional detail to ensure reproducibility. I recommend publication in Nature Communications after the authors address the following points:

1. The authors should clarify the rationale behind the choice of 12.5% surface iodide substitution by phosphite in the DFT calculations shown in Figure 1e.
2. More detailed information is required for the SPV measurements in Figure 1f. Specifically, the WF values under dark and illumination conditions? What is the sample structure—HTL/perovskite/C60 or ITO/HTL/perovskite/C60? What light intensity was used, and is it sufficient to induce flat banding at the perovskite surface? Additionally, can the authors provide the WF values under dark and illumination conditions for samples without the C60 layer?
3. In line 168, the authors claim that the increased SPV amplitude suggests a higher electron concentration in the C60 layer, implying that more electrons are present in the C60 of the pFBPA sample. However, in line 177, they state that n-doping is reduced in the pFBPA sample. Is there a contradiction between these two statements? Could the authors also add the conduction band alignment in Figure S4 for clarity? Furthermore, the UPS spectra seem to be missing—was C60 included in the measured sample structure?
4. What is the chemical state of PCI on the surface? Is there any interaction between PCI and SP? Does the deposition sequence of IP and SP affect device performance?
5. One highlight of this work is the demonstration of large-area perovskite/silicon tandem devices. However, the claim of uniform passivation over large areas is only supported by PL mapping on a 2.5 cm × 2.5 cm sample (Figure 2B and 2C, line 240). Given that the authors employed spin-coating (with solution concentration scaled by ×10), how can uniformity over the 60 cm² area be ensured, especially considering the challenges posed by the anti-solvent method and solution deposition of passivation materials? Increasing concentration of ten compromises film uniformity—could the authors explain why this is not the case here? Also, does the bimolecular approach offer specific advantages for improving long-term stability? More fabrication details for the large-area devices would be beneficial.
6. Since the 1.55 eV perovskite sub-cell yields only 24.75% efficiency (line 320), which is significantly lower than state-of-the-art single-junction devices, does this imply that the effectiveness of the bimolecular approach is limited in MBG perovskites? A discussion of this point would be helpful.
7. For better clarity, the EQE curves in Figure 3d should include the summed EQE of the top and bottom cells, along with the reflectance spectrum.
8. In Figure S27, the authors should include the full certification report. If the reported value is not for the total area, please specify the picture of the mask used.
9. Given that large-area tandem devices are still underexplored, the authors are encouraged to discuss the variation of photovoltaic parameters with increasing device area. For instance, in this work, when scaling from 1 cm² (J_{sc} = 20.24 mA/cm², FF = 81.96%, Voc = 1.983 V, Figure 3A) to 4 cm² (J_{sc} = 18.56 mA/cm², FF = 84.78%, Voc = 1.95 V, Figure S25) and further to 60 cm² (J_{sc} = 19.18 mA/cm², FF = 79.2%, Voc = 1.97 V, Figure 3F), the trends in these parameters merit further

discussion.

Reviewer #2

(Remarks to the Author)

Comments:

In this manuscript, the authors demonstrate homogeneously passivated devices reaching certified efficiencies of 28.9% for an active area of 60 cm², the efficiency of large area perovskite/silicon tandem solar cell is impressive. There is no doubt that SIP treatment can effectively reduce the non-radiative recombination losses between perovskite and C60 and improve the efficiency of perovskite/silicon tandem solar cells, while the advantage for scalability has not been rigorously supported by evidence. So, there are still some comments on scaling up the area of perovskite/silicon tandem solar cells. If the following comments can be answered clearly, I would recommend its publication in Nature Communications.

- #1. In this manuscript, the most attractive result is the high efficiency large area perovskite/silicon tandem solar cells. However, the key factor limiting the efficiency loss during active area scaling of perovskite/silicon tandem solar cells appears not to originate from the perovskite/C60 interface. What distinct scientific challenges arise at the C60/perovskite interface when scaling up solar cell area from 1 cm² to 60 cm²?
- #2. In the introduction, the authors mentioned that evaporated dielectric interlayers (LiF, MgF_x) or ALD-deposited AlO_x can suppress recombination between perovskite and C60 (interface passivation). Evaporation and atomic layer deposition (ALD) are recognized as two primary scalable methods for uniform material fabrication. What advantages does the SIP treatment proposed in this work offer over these techniques in large-area solar cell manufacturing?
- #3. In Figure 1B, the MPA samples show much non-radiative losses than control samples, in Figure 1G, the MPA samples show Voc improvement. The authors think the improvement (<40 mV) with MPA is attributed to the suppression of Pb⁰ defects, which act as deep trap centers in the presence of C60. While the question is PLQY measurement can show the non-radiative losses of the whole film. If Pb⁰ defects on the surface can be passivated by MPA, the PLQY of perovskite film after deposition of C60 will can experimentally validate this hypothesis.
- #4. In Figure S3, the FBPA samples show obvious current density gain than BPA and pFBPA samples, what accounts for this discrepancy?
- #5. Figure 2A and 2B seems inconsistent with each other, in Figure 2B, only control samples suffer from more NR losses with C60/SnO_x. It is commonly observed that non-radiative recombination increases after C60 evaporation. Therefore, the data presented in Figure 2B may indicate a potential inconsistency. The author should check the data in Figure 2B.
- #6. The GIWAXs data in Figure S10 seems asymmetry. The sample alignment might have deviated from the ideal horizontal plane during GIWAXS data collection, affecting scattering vector accuracy, or the GIWAXS data should be calibrated.
- #7. The author should provide more details on the fabrication protocols for large-area perovskite/silicon tandem solar cells. Were both the functional layers and light-absorbing layers in the large-area tandem devices fabricated via spin-coating? If so, how do the spin-coating parameters differ from those used in 1 cm² devices? Additionally, is the annealing process consistent between the two scales?
- #8. In Figure S30 and Figure S33, it is obvious that SIP samples decay obviously during MPP tracking. Which passivators are responsible for this observed phenomenon, pFBPA or PCI? Furthermore, what causes the performance enhancement of the control devices under light illumination? The assertion that Pb⁰ at the interface is the primary factor limiting device efficiency appears inconsistent with this observation.
- #9. It is claimed that the calibration of the sun monitor is checked with three different certified cells with different spectral responses to minimize the spectral mismatch of the sources. What is the spectral response of the three calibrated cells, and how to adjust the spectral mismatch with regard to different type or active area of the tandem cells? Also, the corresponding mismatch factor for sub-cells should be provided.
- #10. Since the IV testing system and stability testing system are different, the spectrum is suggested to be provided, respectively.
- #11. For the large-area testing, it is claimed the shading due to probe bars is corrected after the measurements by application of the correction factor (x1.016) to J_{sc}. The author should provide that how to obtain the value 1.06. Also, the final results of the large-area tandem solar cells need to be corrected regarding the FF variation when increase the light intensity to coordinate the final J_{sc} value.
- #12. Considering the fine stability of the cell, the author should explain the gap between home testing and the 3rd party testing.
- #13. The authors are advised to verify whether the data for the distance of the Cl ... C60 contact in Table S1 is missing or incomplete.
- #14. It is recommended that the authors provide SEM images of the double-side sub-micron textures for the 60 cm² champion device to better illustrate the structural features.
- #15. There is still a significant efficiency loss when scaling the tandem device size from 1 cm² to 60 cm². Could the authors perform an analysis to identify the sources of this efficiency loss?
- #16. The difference in J_{sc} between the 1 cm² and 60 cm² devices is significantly higher than the J_{sc} difference caused by shadowing losses. Could the authors provide an analysis of the mechanisms contributing to this J_{sc} loss?

Reviewer #3

(Remarks to the Author)

The manuscript presents an important advance in scaling up the perovskite silicon tandem solar cells, achieving 28.9 % certified efficiency over 60 cm² through a homogeneous dual passivation strategy (pFBPA and PCI). The approach combining phosphonic acid surface passivation and piperazinium chloride interface passivation is convincing and well supported by several characterization and DFT calculations. The study is thorough and the results are of high relevance to

the field. I recommend this manuscript for publication after the authors address the following clarifications.

Comments and requests for clarification:

1. The manuscript primarily focuses on steady state PL data. To better understand charge extraction and recombination dynamics at the perovskite/C₆₀ interface, could the authors provide transient PL (TRPL) decay curves for the different passivation conditions, both with and without the C₆₀/SnO_x layers?
2. Regarding stability, could the authors provide statistical data (e.g., number of tested samples, distribution of T₈₀ lifetimes) to confirm that the observed degradation trends under operational conditions are representative and reproducible?
3. In large-area devices, how does the 130 °C Ag curing process impact the structural or chemical integrity of the passivation layer? Was there any indication of degradation or performance loss associated with this step?
4. Is there any experimental evidence (e.g., UPS shifts, XPS coverage estimates) that provides insight into the molecular coverage or uniformity achieved by the pFBPA and PCI treatments?
5. Please Include the figures of merit (Voc, Jsc and FF) extracted during the ageing studies to complement the efficiency data.

Version 1:

Reviewer comments:

Reviewer #1

(Remarks to the Author)

The revised manuscript shows significantly improved quality, and I support the publication of this paper as is.

Reviewer #2

(Remarks to the Author)

The authors had revised the manuscript and answered the comments #1-#10 nicely, I don't have any other comments, however, for #11 and #12 I have two more comments.

#11 , The authour claims using our equivalent circuit modeling for tandems (if required, we can provide more details). We would appreciate the author giving more details about the modeling and specify the data sources. (Data supported by published literature or industry consensus)

#12 , Did the authors account for potential inconsistencies in testing due to device degradation? If so, please specify the exact data variations caused by degradation. If not, please elaborate on how differences in the test setup (masks, probes, and so on.) may lead to discrepancies in electrical parameters and explain the underlying reasons

Reviewer #3

(Remarks to the Author)

I have reviewed the revised manuscript and the authors response to my previous comments. The manuscript has improved significantly, and all my concerns have been adequately addressed. I recommend acceptance and publication of the manuscript in its current form.

Version 2:

Reviewer comments:

Reviewer #2

(Remarks to the Author)

The authors have improved great a lot according to the given comments and suggestions. I support the publication of its current version.

We thank the reviewers for their valuable comments. Indeed many of them helped us improve the quality of the manuscript. We answer their comments in red below.

Reviewer #1 (Remarks to the Author)

The authors propose a bimolecular passivation strategy to achieve both field-effect and chemical passivation at the C₆₀ interface. A thorough explanation of the underlying mechanism is provided. As a result, the perovskite/silicon tandem solar cells, particularly the 4 cm² and 60 cm² devices, reach impressive stabilized power conversion efficiencies of 30.74% and 28.9% (certified), respectively. This work is of clear significance to the field of perovskite/silicon tandems, especially with regard to large-area scalable fabrication. Most of the authors' conclusions are well supported by the data, including DFT calculations, SPV measurements, PL mapping, and photovoltaic device performance. The methodology is generally sound and well described. However, certain aspects—particularly related to large-area device fabrication—would benefit from additional detail to ensure reproducibility. I recommend publication in Nature Communications after the authors address the following points:

We would like to thank the reviewer for the valuable comments and feedback and for pointing out possible concerns. In the following, we addressed these points to the best of our knowledge.

1. The authors should clarify the rationale behind the choice of 12.5% surface iodide substitution by phosphite in the DFT calculations shown in Figure 1e.

The rationale behind the choice of 12.5% surface substitution is to demonstrate partial coverage of the surface with the molecules utilized and also to compare the binding energies in full and partial substitution scenarios. Similar bilayer interface passivation strategies with partial contacting have been demonstrated before by Liu et al. using LiF and EDAI₂ (www.nature.com/articles/s41586-024-07997-7). With the low concentrations utilized in this work (e.g., 1 mM), it is likely that we have only partially covered surfaces, and experimentally quantifying such coverage is rather challenging. Hence, in DFT calculations, we considered two possibilities as 12.5% and 100% surface substitution for the pFBPA.

2. More detailed information is required for the SPV measurements in Figure 1f. Specifically, the WF values under dark and illumination conditions? What is the sample structure—HTL/perovskite/C60 or ITO/HTL/perovskite/C60? What light intensity was used, and is it sufficient to induce flat banding at the perovskite surface? Additionally, can the authors provide the WF values under dark and illumination conditions for samples without the C60 layer?

We thank the reviewer for the comment and the questions. The sample structure for the Fig.1(f) is ITO/HTL/perovskite/phosphonic-acid/C60. The light (white LED light) intensity in these measurements is ca. 1-sun (calibrated with a Si diode). Zu et al. (DOI:10.1021/acsami.9b05293) (using the same system of KP used in this work in Potsdam University), varied the light intensity from 0 up to 1.5 sun. It is mentioned that at 1.5 sun, they can assure flat band conditions. However, in Fig.S6 of the article, it has been shown that the WF does not change as the light intensity increases further above 0.61 sun. Even at 0.24 sun,

most of the SPV (~95% of the SPV) has already been reached. Likewise, Gutierrez-Partida (DOI: 10.1021/acsami.4c14194), working with the same setup, and perovskites with a very similar composition and bandgap, demonstrated that with 1-sun illumination, flat banding is satisfied.

Below we show the complete results of WF values under dark and light for ITO/HTL/perovskite/PA/C60 and ITO/HTL/perovskite/PA; SPV is lower in the absence of C₆₀, where similar observations have been shown by Gutierrez-Partida et al. (DOI:10.1021/acsami.4c14194) with increasing SPV after ETL.

Figure R1.KP measurement under dark and light for ITO/HTL/perovskite/PA (top graph) and ITO/HTL/perovskite/PA/C60 (bottom graph) samples (Ref-grey, MPA-red, BPA-blue, FPBA-green, pFBPA-violet, pCl-yellow, PFBPA-PCI – light blue)We updated Fig.S(15) according to the suggestions by the reviewer. We updated the Fig.1(f) captions and SI details for KP measurements for better clarity.

3. In line 168, the authors claim that the increased SPV amplitude suggests a higher electron concentration in the C₆₀ layer, implying that more electrons are present in the C₆₀ of the Pfbpa sample. However, in line 177, they state that n-doping is reduced in the Pfbpa sample. Is there a contradiction between these two statements? Could the authors also add the conduction band alignment in Figure S4 for clarity? Furthermore, the UPS spectra seem to be missing—was C₆₀ included in the measured sample structure?

All BPA-derivatives (BPA, FBPA, and pFBPA) passivate the perovskite/C₆₀ interface, not necessarily the perovskite surface, though. Only samples with fluorinated-BPAs also passivate surface defects. Hence, especially pFBPA achieves the highest SPV amplitude and V_{OC} because both the perovskite surface is passivated (e.g., increasing PLQY upon deposition of pFBPA) and also the losses at the perovskite/C₆₀ interface are minimized. The SPV amplitude (which the original signal is negative since p-i-n devices are measured from the n-side where electrons are accumulated) is proportional to the electron concentration, and any improvement in surface and interface passivation contributes to such an increase. However, **Fig.S4** shows that with pFBPA band energetics are not improved which is the likely causing a stagnation of the FF and minor non-radiative losses at the ETL-interface, **Fig.2(a)**.

To achieve a perfect interface, also band-energetics between perovskite and C₆₀ need to be improved (e.g., by modulating n-type doping at the interface, minimizing the ECM-offset, maximizing EVM offset). With the introduction of pFBPA, there is an increase in the EVM offset, which in fact can minimize the minority carrier concentration (holes at this interface), decreasing recombination. However, there is also a reduction in the surface n-type behavior, which can be read as an increase in minority carrier concentration; however, we believe it is an absence of Pb⁰ that is responsible for this apparent “de-doping” because pFBPA suppresses its formation (see e.g., Chin et al. Science 2023). So, pFBPA passivates defects, but does not provide any improvement in the band energetics that still cause interface recombination losses. This is then further improved with the introduction of PCl. Thus, we believe the statements in lines 168 and 177 are not contradictory, because the decreasing n-type surface behavior is likely a “removal” of Pb⁰ and overall compensated by the increasing EVM offset, and the carrier concentrations increase (compared to the control device without any PA passivation) due to surface and interface passivation.

In **Fig.S(4-16)**, we tried to avoid adding conduction band minima (ECM) via optical bandgap because such analysis will assume a fixed bandgap for each condition and C₆₀, which is not necessarily true. We contacted several well-known groups to perform IPES on these samples, but they were all unavailable, but we agree this would have helped to even better elucidate the energetics.

Samples with C₆₀ are included in the UPS analysis related to **Fig.S16** and **Fig.2(d)** but not in the analysis for the PA-family. The spectra with C₆₀ are present in **Fig.S14** (right side of SECO – high WF and left side of EVM, high ECM-WF offset), and indicated in **Fig.R2**.

Figure R2. SECO and EVM, where circles indicate the measurements on C60.

4. What is the chemical state of PCI on the surface? Is there any interaction between PCI and SP? Does the deposition sequence of IP and SP affect device performance?

We did not conduct any characterization to investigate the interaction between PCI and SP (pFBPA). However, we expect that due to the characteristics (e.g., pFBPA acting as Lewis-acid and PCI as Lewis-base), these molecules would form a salt by protonation reactions ($\text{pFBPA}^+ \text{PCI}^-$) by entering a Lewis acid-base reaction. If needed, we can provide NMR characterization for PCI, pFBPA and pFBPA+PCI.

In this work, we only investigated the following deposition sequence for;

1. pFBPA deposition and annealing, followed by PCI deposition and annealing
2. PFBPA as additive in perovskite ink that accumulates at surfaces and then PCI deposition and annealing
3. PFBPA and PCI mixed (1:1 in IPA) deposition and annealing

These three conditions work (see **Fig.R3**) because pFBPA is directly in contact with the perovskite, where surface passivation can be achieved. Hence, we deem it likely that if PCI is deposited first, the perovskite surface will not be as effectively available for pFBPA surface passivation, lowering the performance.

Figure R3. Open-circuit voltage of single junction devices with different SIP applications.

5. One highlight of this work is the demonstration of large-area perovskite/silicon tandem devices. However, the claim of uniform passivation over large areas is only supported by PL mapping on a $2.5 \text{ cm} \times 2.5 \text{ cm}$ sample (Figure 2B and 2C, line 240). Given that the authors employed spin-coating (with solution concentration scaled by $\times 10$), how can uniformity over the 60 cm^2 area be ensured, especially considering the challenges posed by the anti-solvent method and solution deposition of passivation materials? Increasing concentration of ten compromises film uniformity—could the authors explain why this is not the case here? Also, does the bimolecular approach offer specific advantages for improving long-term stability? More fabrication details for the large-area devices would be beneficial.

In addition to **Fig.2(a)**, we included the PL mapping of the top-cell for a 60 cm^2 device with SIP passivation in **Fig.3(h)** which shows homogeneous passivation across this area. In fact, this is a testimony of homogeneity not just of one layer, but of multiple (e.g., SAM, perovskite, SIP), which would otherwise lead to different forms of inhomogeneities. The major decrease in the PL comes from the perovskite/ C_{60} interface in inverted architecture; hence, any inhomogeneity after SIP deposition would result in low PL areas, which is not the case.

Indeed, the antisolvent method can result in challenges in terms of scaling up, but with an optimized process (for which we included more details in the SI regarding processing of the large-area devices), still in 4-inch samples, we observe a high yield (**Fig.3(e)**).

In terms of shelf and humidity stability, samples with SIP show advantages compared to control devices without passivation, likely due to the hydrophobic pentafluorinated phosphonic acid. However, in terms of operational stability, both pFBPA, PCI, and their combinations accelerate degradation, see **Fig.R4**. Hence, as an important future work, we are investigating alternative surface and interface passivation materials that also provide enhanced stability (e.g., ALD- AlOx in combination with an alternative surface passivation molecule).

Figure R4. Operational stability of single junction devices with different surface treatments

6. Since the 1.55 eV perovskite sub-cell yields only 24.75% efficiency (line 320), which is significantly lower than state-of-the-art single-junction devices, does this imply that the effectiveness of the bimolecular approach is limited in MBG perovskites? A discussion of this point would be helpful.

The MBG device achieved the following performance parameters in this work (current density of 25.01 mA/cm², V_{oc} of 1.18V and reverse scan FF of 83.86% at 0.1 cm²). An example state-of-the-art inverted perovskite solar cell device performance from Chen et al. (DOI:10.1126/science.adm9474) is 1.174V, 26.13 mA/cm² and 85.2%, where a similar perovskite absorber (Cs_{0.05}FA_{0.85}MA_{0.10}PbI₃) is utilized with a slightly lower bandgap (1.53 eV). Hence, one major difference is likely the bandgap that results in slightly higher (+5 mV) V_{oc} and lower (-1.12 mA/cm²) current densities in our case. Moreover, the area of the high-efficiency devices is in the range of 0.05 cm² which in our case 0.1 cm² that can influence the resistive losses at the TCO, lowering FFs. Lastly, unfortunately, we do not have access to the high-haze (i.e., Asahi FTO) textured FTO-glass, which makes a significant difference in improving the current densities due to enhanced light trapping. In this work, we also utilized an FTO from another supplier, but the optical performance lags behind the high-performance FTO by Asahi. Metallization design of the single-junctions can also be improved further to maximize the efficiency.

7. For better clarity, the EQE curves in Figure 3d should include the summed EQE of the top and bottom cells, along with the reflectance spectrum.

We updated the EQE plot in **Fig.3(d)** with the summed EQE of sub-cells. In our custom-built EQE setup, the spot size is smaller than the active area where we can measure the EQEs in between metal fingers. Unfortunately, measuring the reflectance spectrum of such devices with metal fingers is rather challenging with our capabilities and requires assumptions regarding the metallized fraction; hence, we tried to avoid plotting 1-Reflectance. If required, we can find a collaborator with a PV-Tools EQE setup that can measure 1 cm x 1 cm EQE and reflectance, respectively, however we want to highlight that our previous results resulted in minimal differences compared to certification institutes we sent tandems to, reassuring our measurement procedures are correct.

8. In Figure S27, the authors should include the full certification report. If the reported value is not for the total area, please specify the picture of the mask used.

We included the full report here [redacted] and note that the active and the measurement area are the same. Also, the picture of the cell and the mask is included in **Fig.S29** along with MPP results from the certification.

Figure R5. 60 cm² tandem device with a respective IV mask. The active area is defined by the top-TCO (ITO). The masked area and the active area are similar.

Werkskalibrierschein / Proprietary calibration report

Objekt / Object: monocrystal multi-junction solar cell

Hersteller / Manufacturer: Callab PV Cells

Typ / Type: PSC/2H

Fabrikant/Hersteller-Name / Manufacturer: CEMMOG / (damp)

Auftraggeber / Customer: RSC Japan-Deck I, CH-2000 Neuchâtel, Switzerland

Auftragsnummer / Order No.: 112CEM0724

Abzahl der Seiten des Kalibrierscheins / Number of pages of the certificate: 8

Datum der Kalibrierung / Date of calibration: 16.10.2024

Person in charge: Jochen Roth-Eisinger, Astrid Semerari

1. Beschreibung des Kalibriergegenstandes / Description of the calibrated object

Das Messobjekt ist eine Solarzelle, Typ PSC/2H. Die Stabilität der Solarzelle wurde nicht untersucht. The object under test is a PSC/2H solar cell. The temporal stability of the solar cell performance was not controlled.

Typ / Sample type: monocrystal multi-junction solar cell

Material: PSC/2H

Fläche / Area [cm²]: 60.74

Füllfaktorenabgleich / Area adjustment: 80 aperture area

W-Kontakt / W-contact: Vorventilator / Front 288

Kontakt / Contact: Rückseite / Rear Fully metallized

4. Messergebnis / Measurement results

Die Messergebnisse beziehen sich ausschließlich auf das oben genannte Messobjekt. These results exclusively refer to the measurement object above.

Kennwertparameter des Messobjektes unter Standardtestbedingungen (STC) / Inverse parameter under Standard Test Conditions (STC):

Vorwärtsrichtung / Forward scan direction	Rückwärtsrichtung / Reverse scan direction	steady state MPPT
V_{oc} = 1980 ± 20 mV	V_{oc} = 1980 ± 20 mV	
$I_{sc}(25.2 \pm 0.06)^\circ$ = 1148 ± 20 mA	I_{sc} = 1148 ± 20 mA	
I_{mp} = 1077 mA	I_{mp} = 1075 mA	I_{mp} = 1072 ± 40 mA
P_{max} = 1835 mW	P_{max} = 1829 mW	P_{max} = 1837 ± 35 mW
V_{mp} = 1791 mV	V_{mp} = 1791 mV	V_{mp} = 1792 ± 30 mV
FF = 81.2 %	FF = 81.8 %	FF = 81.9 ± 1.4 %

2. Messverfahren / Measurement procedure

Die Kalibrierung des Kalibrierobjektes wird gemäß [1] mit einem Zweilampen-DC-Sonnensimulator durchgeführt. Die Einstrahlung wird mit Hilfe einer Monochromator während der gesamten Messdauer aufgetrennt und dem Schwankungen bezüglich der Messung linear korrigiert. Die Divergenz der Strahlung ist < 1°. Die Solarzelle wird auf einem Vakuumprobentisch thermisch stabilisiert. The calibration of the test sample was performed at Standard Test Conditions (STC) with a dual light steady-state solar simulator according to [1]. The irradiance is controlled with a monitor cell during the measurement in order to correct fluctuations linear. The divergence of the peripheral beams is < 1°. The solar cell is kept at a constant

Rückführung der Referenzmesswerte / Traceability of the reference solar cell

Identifikations-Nr. / Identity No.	Kalibrierung / Traceability
10010000	Callab PV Cells
10010000	PTB
10010000	PTB

Die Korrektur der spektralen Fehlerleistung (Mischlicht), die durch die Abweichung der spektralen Verteilung des Sonnen-Simulators vom Standard-Spektrum AM1.5G [2] in Kombination mit den verschiedenen spektralen Empfindlichkeiten von Referenzzelle und Messobjekt entsteht [4], wurde durch eine erweiterte Mischlichtkorrektur [5], wie in [2] beschrieben, korrigiert. Dazu wurde die spektrale Verteilung der Bestrahlung (Sonnensimulator) mit einem Spektrodiometer und die spektrale Empfindlichkeit des Messobjektes mit einem abgestimmten Messobjekt [4] gemessen. The spectral mismatch is caused by the deviation of the simulator spectrum from the standard spectrum AM1.5G [2] in combination with the difference between the spectral response of the reference cell and that of the device under test (DUT) - it is calculated by a generalized mismatch correction [4] as described in [2]. For the spectral mismatch correction the spectral distribution of the solar simulator is measured with a spectroradiometer, the spectral response of the DUT is measured with a pump-probe setup according to [5]. Der P_{max} wurde durch MPPT-Tracking über 300s bestimmt. Die angegebene P_{max} ist der Mittelwert (Dahbereich 1 s.) dieser stabilisierten Messung. Inzwischen wurde die IV-Kennlinie in zwei Richtungen ($I_{sc} \rightarrow I_{mp}$ und $I_{mp} \rightarrow V_{oc}$) aufgenommen. The P_{max} was determined by MPPT-tracking for 300s. The reported P_{max} represents the average value of the range (one range 1 s.) of this stabilized measurement. Afterward, the I-V curve was determined with a scan in both directions ($I_{sc} \rightarrow I_{mp}$ and $I_{mp} \rightarrow V_{oc}$).

Empfindlichkeit / Spectral response	9001112CEM0724	Dahbereich / Time range:	MS - 781 s
----------------	--------------------------	------------

Die Rückführung der Spektralstrahlung auf Si-Einheiten erfolgte über den Vergleich mit einer Standardlampe. The traceability of the measurement of the spectral distribution to Si-units is achieved using a standard lamp for the calibration of the spectroradiometer.

Identifikations-Nr. / Identity No.	Kalibrierung / Traceability
EN-502-450	Callab PV Cells
EN-502-450	PTB

5. Zusatzinformationen / Additional information

Fläche / Area (app): A = 60.74 ± 0.12 cm²

3. Messbedingungen / Measurement conditions

Standardtestbedingungen (STC) / Standard Test Conditions (STC):

Absolute Bestrahlungsstärke / Total irradiance: 1000 W/m²

Nominalwert der Temperatur des Messobjektes / Nominal Value of Temperature of the DUT: 25 °C

Spektrale Bestrahlungsverteilung / Spectral irradiance distribution: AM1.5G G4 (2019) [6]

Umgebungsbedingungen / Ambient conditions:

Temperatur / Temperature: (23 ± 1) °C

Luftfeuchtigkeit / Humidity: (50 ± 5) %

Die Messung der IV-Kennlinie (Strom-Spannung-Kennlinie) des Messobjektes erfolgt mit Hilfe eines Vierquadranten-Netztes und eines Kalibrierwiderstandes. Die Temperatur der Solarzelle wird mit einem Temperatursensor ermittelt und auf (25±0,1)°C eingestellt. The measurement of the I-V curve is performed with a four-quadrant power amplifier and a calibration resistor. The temperature of the solar cell is determined by a sensor and adjusted to (25±0,1)°C.

Kontaktierung der Bestrahlungseite / Contacting of illuminated side	288, 8m	Bestrahlungseite / Illuminated side	Frontseite
Rückseite / Backside	288, 8m, fac.	Temperatursensor / Temperature sensor	Temperaturtastloch PT 100

8m: busbar resistance neglecting
 gm: grid resistance neglecting
 fm: fully metallized

fac: full area contact
 fac: local contact
 nrc: non-reflective chuck
 hrc: highly reflective chuck

Die Art der Kontaktierung der Solarzelle und die Reflexivität des Messobjektes können Einfluss auf das Messergebnis haben. Eine Abweichung von den hier beschriebenen Messbedingungen führt ggf. zu anderen Messwerten. Changes to the measurement conditions, such as the type of contacting or the reflectivity of the chuck, can lead to deviations which influence the measurement results.

[1] [2] in total area (app) in aperture area, [A] in designated illumination area [7].

Angaben mit jeweils die erweiterte Messunsicherheit, die sich aus der Standardmessunsicherheit durch Multiplikation mit dem Faktor k=2 ergibt. Sie wurde gemäß dem "Guide to the expression of Uncertainty in Measurement" ermittelt. Sie entspricht bei einer Normalverteilung der Abweichungen vom Messwert einer Überdeckungswahrscheinlichkeit von 95%. The expanded measurement uncertainty resulting from the standard measurement uncertainty multiplied with a factor k=2 is specified. The calculation was carried out according to the "Guide to the expression of Uncertainty in Measurement". The value corresponds to a Gaussian distribution denoting the deviations of the measurement value within a probability of 95%.

Figure R6. Full certification report from Fraunhofer ISE Callab.

9. Given that large-area tandem devices are still underexplored, the authors are encouraged to discuss the variation of photovoltaic parameters with increasing device area. For instance, in this work, when scaling from 1 cm² (Jsc = 20.24 mA/cm², FF = 81.96%, Voc = 1.983 V, Figure 3A) to 4 cm² (Jsc = 18.56 mA/cm², FF = 84.78%, Voc = 1.95 V, Figure S25) and further to 60 cm² (Jsc = 19.18 mA/cm², FF = 79.2%, Voc = 1.97 V, Figure 3F), the trends in these parameters merit further discussion.

There are multiple differences in the fabrication of 1 cm² (on 2.5 cm x 2.5 cm substrates) vs. 4 and 60 cm² (fabricated on 4-inch wafers) that cause differences in the device performance.

1. In large-area devices, to ensure conformal coverage and homogeneity of the top-electrode stack over larger areas, we use a slightly thicker C60-layer (e.g., 25 nm C60). For the deposition of ALD-SnOx, another ALD tool with a thinner baseline recipe (125

cycles from Oxford ALD) compared to a 175-cycle baseline for 1 cm² devices from PICOSUN ALD. Lastly, 45 nm ITO is utilized. Increasing thicknesses in such layers increase parasitic absorption losses (e.g., estimated loss of 0.6 mA/cm²).

2. In 1 cm² devices, 30-35 nm of highly-transparent IZrO (with as-deposited mobilities reaching 40 cm²/V s) is utilized as top-TCO. However, for the large area device, ITO (45 nm) is utilized. On one hand, the main reasons for this change are the low contact resistance between the screen-printed grid and the ITO and the deposition capabilities (in an industrial ITO sputtering tool we can deposit on 8 x 4-inch wafer at a time, and for IZrO we can only deposit on 1 wafer at a time). On the other hand, as-deposited ITO is less transparent than as deposited IZrO, causing increased parasitic absorption.
3. During the screen-printing process, the samples are annealed at 130°C for 10 min in ambient. However, for the baseline absorber utilized in 1cm², this temperature is too high, which causes degradation. We identified the decomposition of the absorber due to excess PbX₂ content as the main driver. Hence, to withstand these temperatures, the PbX₂ excess percentage is decreased from 6% to 3%, which decreases the device performance by approximately 0.5 abs % due to 10-15 mV lower V_{OC}'s due to decreasing absorber quality.
4. Another major difference is the bottom-cell architecture utilized in 1cm² cells. Specifically, the 1cm² bottom-cells (front-flat rear textured architecture) are fabricated at EPFL with tailored PECVD and protective layers including (nc-Si(n), nc-Si(p), nc-SiOx(n), back-reflector SiOx-np, back-protection SiOx, ... [www.cell.com/joule/fulltext/S2542-4351\(24\)00199-5](http://www.cell.com/joule/fulltext/S2542-4351(24)00199-5) and DOI: 10.1002/solr.202400704) with lifetimes reaching 5-6 ms at 5e10¹⁵ cm⁻³ carrier concentration (high-injection), close to values reported by Longi in their latest publication, <https://www.nature.com/articles/s41586-024-07997-7>. Fabrication of such bottom-cells includes many additional deposition steps, which are not viable for the fabrication of wafer-scale devices weekly. However, bottom cells larger than 1cm² are fabricated at CSEM (either double-side nanotextured or front-flat rear textured) with PECVD layers only based on a-Si (e.g., a-Si (p), a-Si (n), i-a-Si) without any additional protective layers or back-reflector. The lifetime of such industrially relevant bottom-cells from CSEM is 2-2.5 ms at 5e10¹⁵ cm⁻³. Hence, the optoelectronic performance of 1cm² vs. larger bottom cells is unfortunately different due to unavailability and impracticality. Reference 1 cm² devices fabricated with the industrially relevant CSEM bottom cells on average, provide 31.5% efficiency, which is only 1.4 abs% more than the in-house 60 cm² results (30.1% champion).
5. Moreover, in a 1cm² device, the pFFs of each sub-cell are relatively high and similar (e.g., 85-86%), which makes the optimum current mismatch condition close to current matching. However, for the 60 cm² devices, due to challenges in scaling up the perovskite sub-cell, the pFF of the top-cell is lower than the bottom-cell. Such asymmetry in pFF moves the optimum efficiency point towards a bottom-limited scenario. Hence, we observed higher FF's and PCE's when the bottom-cell limitation increases (e.g., when J_{sc} of the top-cell increases and J_{sc} of the bottom-cell decreases). Hence, for 60 cm² devices, we increased the perovskite thickness (by modifying the spin recipe, see SI), as shown in **Fig.R(17)** from the lower response of the Si sub-cell of the 60 cm² device in the wavelength range 600 nm to 750 nm.

- Based on a loss-estimation we think that, by utilizing a back-reflector and optimized PECVD films and protective layers, a 60 cm² tandem device with the top cell demonstrated in this work can achieve: VOC= 1.98V, FF = 80%, and JSC = 19.75 mA/cm² – PCE = 31.28%.

Lastly, the number of fabricated 1 cm² devices reaches 250 in this work, however, for the 60 cm² devices, this number is only around 20, and for 4 cm², 14. Hence, the effort demonstrated and the throughput are significantly different. Larger devices are only at the beginning of their learning curve.

Here we plot the comparison of in-house champion measurements for different areas, see Fig.R7. The top-cell of the 4 cm² device has a lower bandgap (1.63 eV compared to 1.65 eV), which results in lower current densities in the limiting bottom-cells (lower Jsc), slightly lower Voc, but significantly boosted FF due to increasing mismatch towards Si sub-cell. Since there are slight differences, we think the readers will not benefit from further speculations.

Figure R7. PV performance parameters of champion devices with different areas.

Reviewer #2 (Remarks to the Author)

Comments:

In this manuscript, the authors demonstrate homogeneously passivated devices reaching certified efficiencies of 28.9% for an active area of 60 cm², the efficiency of large area perovskite/silicon tandem solar cell is impressive. There is no doubt that SIP treatment can effectively reduce the non-radiative recombination losses between perovskite and C60 and improve the efficiency of perovskite/silicon tandem solar cells, while the advantage for scalability has not been rigorously supported by evidence. So, there are still some comments on scaling up the area of perovskite/silicon tandem solar cells. If the following comments can be answered clearly, I would recommend its publication in Nature Communications.

We would like to thank the reviewer for the valuable comments and feedback and for pointing out possible concerns. In the following, we addressed these points to the best of our knowledge. Moreover, changes are highlighted in the revised manuscript.

#1. In this manuscript, the most attractive result is the high efficiency large area perovskite/silicon tandem solar cells. However, the key factor limiting the efficiency loss during active area scaling of perovskite/silicon tandem solar cells appears not to originate from the perovskite/C60 interface. What distinct scientific challenges arise at the C60/perovskite interface when scaling up solar cell area from 1 cm² to 60 cm²?

We believe that the main challenges in the scaling of perovskite-Si tandems are the homogeneous deposition of both the high-quality absorber and the dual passivation agents over larger areas. Moreover, the main performance-limiting interface in inverted solar cells after the demonstration of SAMs (e.g., 2PACz and Me-4PACz) is the perovskite-C₆₀ interface. In small area devices (e.g., 1 cm² or lower), many passivation strategies have been demonstrated to be effective (e.g., LiF, ALD-AlO_x, PDAI₂, PI, ...); however, preserving such effectiveness in large devices hasn't been shown yet for perovskite-Si tandems. Even though evaporated LiF, MgF_x, and ALD-AlO_x are deposited from techniques that allow a priori conformal deposition, the process window due to their insulating nature is extremely small (e.g., between 8-12 cycles for ALD-AlO_x – DOI: 10.1002/adma.202311745). Despite smart solutions to this problem exist (e.g., patterning LiF, DOI 10.1002/aenm.202302132), depositing such thin insulating films with sub-nm precision is rather challenging for larger devices. In that regard, solution-processed surface and interface passivation strategies (e.g., pFBPA&PCI) that preserve their homogeneity across larger areas despite being deposited via solution processing are attractive alternatives.

#2. In the introduction, the authors mentioned that evaporated dielectric interlayers (LiF, MgF_x) or ALD-deposited AlO_x can suppress recombination between perovskite and C60 (interface passivation). Evaporation and atomic layer deposition (ALD) are recognized as two primary scalable methods for uniform material fabrication. What advantages does the SIP treatment proposed in this work offer over these techniques in large-area solar cell manufacturing?

We agree with the reviewer that, especially ALD-AIOx, since it is already widely utilized in the PV industry (see **Fig.R8**) and projected to increase its share, is a very promising interface passivation candidate. ALD-AIOx only possesses two disadvantages: Firstly, a small processing window (e.g., works effectively between 8-12 cycles), hence, strong thickness sensitivity. Secondly, it increases resistive losses due to its insulating nature. By patterning ALD-AIOx (see an example proof-of-concept AIOx patterning in **Fig.R9**), we are working on improving the processing window and improving repeatability as part of separate future work. When we compare ALD-AIOx to solution-processed SIP treatment, we can count multiple advantages. SIP treatment has a wider range of processing window (0.5 mM to 2.5 mM) and can be washed with IPA afterwards to remove excess unbound passivation molecules. This contributes to the repeatability of the results demonstrated in this work, and makes the devices more tolerant to errors. The second advantage is that the SIP decreases the resistive losses across the interface, contributing to enhanced FF's. We PCI improves the band energetics charge extraction. Hence, in our lab-scale devices, we can never achieve the efficiency demonstrated by SIP using ALD-AIOx, see **Fig.R10**.

Lastly, ALD-AIOx is grown at low temperature (e.g., 75°C), unlike the Si industry (e.g., 125-150°C) deposition temperatures and post-activation temperatures of 400°C) on the perovskite surface. Significantly lower growth temperatures can influence the homogeneity of this layer because of insufficient thermal energy and surface binding sites for nucleation. Recently, Choi et al., (DOI: 10.1016/j.joule.2024.12.002) tackled this homogeneity problem by AVAI surface modification to provide functional groups and protection to enable 100°C deposition temperature for effective ALD-nucleation. Hence, scaling up ALD-AIOx also requires certain surface modifications. Moreover, we did not find any evidence that ALD-AIOx passivates the perovskite surface, unlike pFBPA and still the perovskite-C60 interface is not loss-free with ALD-AIOx, meaning the achieved open-circuit voltages lag behind the SIP counterparts by at least 20 mV, see **Fig.R10**.

Fig. 24: Market share of technologies for passivation.

Figure R8.ITRPV report 2025, market share of technologies for passivation.

Figure R9. Patterned ALD-AIOx interface passivation example, bright areas are the locations with AIOx in the PL mapping and device results.

Figure R10. Open-circuit voltage of perovskite single junctions with different interface passivation's in combination with different absorbers.

#3. In Figure 1B, the MPA samples show much non-radiative losses than control samples, in Figure 1G, the MPA samples show V_{oc} improvement. The authors think the improvement (<40 mV) with MPA is attributed to the suppression of Pb0 defects, which act as deep trap centers in the presence of C60. While the question is PLQY measurement can show the non-radiative losses of the whole film. If Pb0 defects on the surface can be passivated by MPA,

the PLQY of perovskite film after deposition of C60 will can experimentally validate this hypothesis.

Indeed, after the deposition of MPA, NR-loss increases slightly (5 to 10 mV) compared to the bare perovskite surface, and after completion of the full devices, NR-losses of MPA-treated devices increase further up to 150-155 mV (0.25-0.3% PLQY). However, for the control device without any treatment, these losses increase up to 170-175 mV. Hence, these results show that MPA provides a certain amount of interface passivation compared to the bare-perovskite case. In this case, we hypothesize that passivating Pb⁰ in fact does not necessarily decrease NR-losses since it is not an active deep defect in the absence of C60, however, after completion of the top-stack MPA-treated devices exhibit higher PLQY because such devices do not suffer from Pb⁰-induced interface recombination.

#4. In Figure S3, the FBPA samples show obvious current density gain than BPA and pFBPA samples, what accounts for this discrepancy?

We believe FBPA condition is a statistical outlier; if needed, we can provide a comparison of EQEs between conditions. We do not think the differences are statistically relevant (~1% relative difference). We observe no obvious differences in the EQEs and all the devices fabricated in this work have the same thickness and bandgap. Indeed, we see in TRPL studies (shown below) faster charge extraction to C60 with SIP, but we don't think this translates into higher collection efficiencies in this case. Given reports from literature charge extraction is limited by the HTL, so this should not account for a change in JSC by itself (<https://www.sciencedirect.com/science/article/pii/S2542435121003548>).

Figure R11. Jsc results for single junction devices with different PA treatment

#5. Figure 2A and 2B seems inconsistent with each other, in Figure 2B, only control samples suffer from more NR losses with C60/SnOx. It is commonly observed that non-radiative recombination increases after C60 evaporation. Therefore, the data presented in Figure 2B may indicate a potential inconsistency. The author should check the data in Figure 2B.

We thank the reviewer for raising this point. Indeed, in the PL-imaging results, the samples with C60/SnOx show less NR-losses compared to the bare-perovskite scenario. Upon the

suggestion of Review#3, we conducted TRPL measurements on half-cell with different passivation with and without C60/SnO_x (same set of samples in the PL mapping – see Fig.R12). Also, in the TRPL measurements we saw enhanced lifetimes for the samples with SP/IP/SIP with the presence of ETL-stack (see **Fig.R12(b)**) confirming the observations in the PL-mapping (**Fig.R12(a)**).

Figure 12. A) Non-radiative loss mapping measured using wide-field absolute PL (1cm scale bar) with and without C₆₀/SnO_x for different conditions SP (pFBPA), IP (PCI), and SIP (PCI/pFBPA) on 2.5 x 2.5 cm² substrates. B) Transient PL decay time (t₃) with and without C₆₀/SnO_x for different conditions, SP, IP, and SIP.

We double-checked these measurements and deem it likely that differences arise from the measurement procedures and sample history. For the samples in “previous Fig.2(a)” – PLQY results, we utilized the same substrates, meaning before C₆₀/SnO_x we measured the PLQY with inevitable air exposure (which might have an influence - doi/10.1002/aenm.202501225), and, PLQY of the same sample is measured after C₆₀/SnO_x, which has been done in EPFL within a few hours on the same day. However, for the PL-imaging, a set of samples has been fabricated with and without C₆₀/SnO_x and sent to IPVF. Then, the PL-imaging measurements have been conducted on these different samples. For the TRPL measurements, similar procedure to the PL-imaging is followed and samples (with and without C₆₀/SnO_x) are sent to IPVF. Hence, likely the differences in sample history plays a role in differences in the PL-imaging and TRPL results to the PLQY results (previous Fig.2a). Since, the sample history in the PL-imaging and TRPL is identical, we decided to remove the PLQY (previous Fig.2(a)) and replace it with an excerpt of the TRPL results. Hence, Fig.2 and parts in the main text related to PL mapping and TRPL are updated.

#6.The GIWAXs data in Figure S10 seems asymmetry. The sample alignment might have deviated from the ideal horizontal plane during GIWAXS data collection, affecting scattering vector accuracy, or the GIWAXS data should be calibrated.

We thank the reviewer for raising this point, indeed there is an asymmetry. The GIWAXS data are measured on a scanning instrument (the detector moves at constant distance to the sample) which creates a number of artefacts (also non-linear Q-compression) which cannot be fixed. Finding best compromises in conversion to Q-space (we believe it is not essential to show a 2D-image) can create asymmetry, which is what happened here. Ideally, measurements should have conducted on WAXS beamline, which measures on standard beamline geometry

(orthogonal detector) and converts to Q correctly. However, we do not have the capabilities to conduct such measurements within the next few months and don't think the main results are impacted.

#7. The author should provide more details on the fabrication protocols for large-area perovskite/silicon tandem solar cells. Were both the functional layers and light-absorbing layers in the large-area tandem devices fabricated via spin-coating? If so, how do the spin-coating parameters differ from those used in 1 cm² devices? Additionally, is the annealing process consistent between the two scales?

In the large area devices, the following layers are spin-coated on 4-inch wafers; SAM, SiO_x-np, perovskite, SIP. In the SI, we included more details (e.g., perovskite processing, amounts of solution) regarding the large-area processing of the devices.

#8. In Figure S30 and Figure S33, it is obvious that SIP samples decay obviously during MPP tracking. Which passivators are responsible for this observed phenomenon, pFBPA or PCI? Furthermore, what causes the performance enhancement of the control devices under light illumination? The assertion that Pb⁰ at the interface is the primary factor limiting device efficiency appears inconsistent with this observation.

Both, pFBPA and PCI, when utilized separately, accelerate degradation, see **Fig.R(4)**. When combined, we don't observe any synergetic effects stabilizing this interface. We hypothesize that the control devices initially have imperfect band energetics, but over time, with external stressors, the band energetics improve, contributing to the improved performance. However, the SIP-treated device starts the light-soaking test with a perfectly passivated and aligned state, meaning that upon stressors, when the system is perturbed, these devices degrade. We think the observed improvements in the control devices stem from a lattice relaxation similar to what was observed here, albeit slower: <https://www.science.org/doi/10.1126/science.aap8671>

#9. It is claimed that the calibration of the sun monitor is checked with three different certified cells with different spectral responses to minimize the spectral mismatch of the sources. What is the spectral response of the three calibrated cells, and how to adjust the spectral mismatch with regard to different type or active area of the tandem cells? Also, the corresponding mismatch factor for sub-cells should be provided.

We thank the reviewer for raising this important point and we provided more details in the SI with the following updates, and here we provide the spectrum, Fig.R(13);

Figure R13. Spectrum of the solar simulator used for 1cm² devices.

For the 1cm² devices;

‘Initially, to adjust the intensity of the Xe lamp, a calibrated encapsulated cell (WPVS – blue cell) containing a monocrystalline silicon solar cell, which is covered with a KG5-filter was used. Then to adjust the intensity of Ha lamp, a calibrated encapsulated cell (WPVS – red cell) containing a monocrystalline silicon solar cell, which is covered with a RG780-filter was used (see **Fig.R(14)**). Then, overall spectrum is checked with a calibrated encapsulated cell (WPVS without filter broadband – black cell). To further fine-adjust the intensity of the Ha lamp, a tandem cell within the fabricated batch was first measured by EQE, then the lamp intensity was adjusted to match the integrated EQE of the bottom cell with the measured Jsc, notably with the conditions in which the bottom cell is the current limiting cell. All calibrated Si cells are provided by Fraunhofer ISE.’

For the larger devices;

‘Initially, to adjust the intensity of the Xe lamp, a calibrated encapsulated cell (WPVS – blue cell) containing a monocrystalline silicon solar cell, which is covered with a KG5-filter was used. Then to adjust the intensity of Ha lamp, a calibrated encapsulated cell (WPVS – red cell) containing a monocrystalline silicon solar cell, which is covered with a RG780-filter was used. Then, overall spectrum is checked with a calibrated encapsulated cell (WPVS without filter – black cell). All calibrated Si cells are provided by Fraunhofer ISE.’

Figure R14. Filters for the calibrated cells

The spectral mismatch factor is calculated as 1.009 for the top cell, and 1.000 for the bottom cell (in which case the reference and test cells are both the ones used in the tandems within a specific batch and the EQEs of these test and reference cells are assumed to be identical; we tested this extensively in the past with typically < 1% rel. error across many devices).

#10. Since the IV testing system and stability testing system are different, the spectrum is suggested to be provided, respectively.

The spectrum of the stability testing setup (LED-based and solar simulator) is provided in the **Fig.R(15)**. Indeed, the IV testing and stability setup produce different spectra due to the nature of the light source (LED vs. Xe-Ha). UV response is stronger in the IV setup, and in the stability setup there is dip in the spectrum around 400 nm.

Figure R15. Spectrum of the LED-based stability setup in comparison to AM1.5G and spectrum of Wacom solar simulator to AM1.5G.

#11. For the large-area testing, it is claimed the shading due to probe bars is corrected after the measurements by application of the correction factor (x1.016) to J_{sc} . The author should provide that how to obtain the value 1.06. Also, the final results of the large-area tandem solar cells need to be corrected regarding the FF variation when increase the light intensity to coordinate the final J_{sc} value.

We thank the reviewer for raising this important point. To obtain the correction factor (**x 1.016**), a 60 cm² device is measured with probes on the busbar without shading the device, and the $J_{sc-probe}$ is compared to the $J_{sc-probebar}$ measured with probe bars (which causes shading). Such a comparison yielded the x 1.016 factor. The difference between probe and probe bar measurements, apart from the J_{sc} influenced by the shading, was the FF due to differences in the resistive losses. In our scenario, we measure the devices and only scale the J_{sc} afterwards, and leave FF and V_{oc} untouched. Hence, we don't account for the FF & V_{oc} variation when the light intensity increases to 1.016 sun instead of 1 sun.

Using our equivalent circuit modeling for tandems (if required, we can provide more details), we simulated the influence of increasing the light intensity from 1 sun to 1.016 sun. The changes in the V_{oc} and FF with comparison to the scaling method can be seen in the following table;

Condition	Voc (V)	FF (%)	Jsc (mA/cm²)	PCE (%)
1 – 1 sun	1.960	80.86	19.000	30.110
2 – 1.016 sun	1.961	80.78	19.304	30.579
3 – only Jsc scaled x 1.016	1.960	80.86	19.304	30.594

Hence, due to the minor (0.015 abs % PCE) efficiency difference, we conclude that such a difference is negligible.

#12. Considering the fine stability of the cell, the author should explain the gap between home testing and third-party testing.

Here is a comparison of in-house measurements (19.18 mA/cm², 79.22% and 1.974V – mask area measured 58.4 cm²) and certified results (18.86 mA/cm², 78.2%, 1.965V – mask area measured 60.74 cm²). There is a difference in the mask area, even though we provided the mask, which we measured beforehand. Indeed, we were also interested in understanding this difference, and enquired at the certification lab, but so far, we do not have an answer. It is possible they cut a new mask from the black low-IR reflecting material, because the IR reflectance of rubber masks (the commonly used material for masks in our lab) is not their standard material.

#13. The authors are advised to verify whether the data for the distance of the Cl ... C60 contact in Table S1 is missing or incomplete.

We thank the reviewer for raising this point, and we corrected the caption of the table accordingly.

‘Table S1. Binding energy (E_b , eV), distance of the N/O/F/I...C₆₀ contact (Å), and Bader charge accumulation on C₆₀ for the different perovskite surfaces.’

#14. It is recommended that the authors provide SEM images of the double-side sub-micron textures for the 60 cm² champion device to better illustrate the structural features.

We included this cross-section SEM image in the SI (Fig.S21(b)) to highlight the front nanotexturing of the 60 cm² champion device.

Figure R16. Cross-section SEM image of nanotextured large area tandems (scale bar 500 nm).

#15. There is still a significant efficiency loss when scaling the tandem device size from 1 cm² to 60 cm². Could the authors perform an analysis to identify the sources of this efficiency loss?

There are multiple differences in the fabrication of 1 cm² (on 2.5 cm x 2.5 cm substrates) vs. 4 and 60 cm² (fabricated on 4-inch wafers) that cause differences in the device performance.

7. In large-area devices, to ensure conformal coverage and homogeneity of the top-electrode stack over larger areas, we use a slightly thicker C₆₀-layer (e.g., 25 nm C₆₀.) For the deposition of ALD-SnO_x, another ALD tool with a thinner baseline recipe (125 cycles from Oxford ALD) compared to a 175-cycle baseline for 1 cm² devices from PICOSUN ALD. Lastly, 45 nm ITO is utilized. Increasing thicknesses in such layers increase parasitic absorption losses (e.g., estimated loss of 0.6 mA/cm²).
8. In 1 cm² devices, 30-35 nm of highly-transparent IZrO (with as-deposited mobilities reaching 40 cm²/V s) is utilized as top-TCO. However, for the large area device, ITO (45 nm) is utilized. On one hand, the main reasons for this change are the low contact resistance between the screen-printed grid and the ITO and the deposition capabilities (in an industrial ITO sputtering tool we can deposit on 8 x 4-inch wafer at a time, and for IZrO we can only deposit on 1 wafer at a time). On the other hand, as-deposited ITO is less transparent than IZrO, causing increased parasitic absorption.
9. During the screen-printing process, the samples are annealed at 130°C for 10 min in ambient. However, for the baseline absorber utilized in 1cm², this temperature is too high, which causes degradation. We identified the decomposition of the absorber due

to excess PbX₂ content as the main driver. Hence, to withstand these temperatures, the PbX₂ excess percentage is decreased from 6% to 3%, which decreases the device performance by approximately 0.5 abs % due to 10-15 mV lower Voc's due to decreasing absorber quality.

10. Another major difference is the bottom-cell architecture utilized in 1cm² cells. Specifically, the 1cm² bottom-cells (front-flat rear textured architecture) are fabricated at EPFL with tailored PECVD and protective layers including (nc-Si(n), nc-Si(p), nc-SiOx(n), back-reflector SiOx-np, back-protection SiOx, ... [www.cell.com/joule/fulltext/S2542-4351\(24\)00199-5](http://www.cell.com/joule/fulltext/S2542-4351(24)00199-5) and DOI: 10.1002/solr.202400704) with lifetimes reaching 5-6 ms at 5e10¹⁵ cm⁻³ carrier concentration (high-injection), close to values reported by Longi in their latest publication, <https://www.nature.com/articles/s41586-024-07997-7>. Fabrication of such bottom-cells includes many additional deposition steps, which are not viable for the fabrication of wafer-scale devices weekly. However, bottom cells larger than 1cm² are fabricated at CSEM (either double-side nanotextured or front-flat rear textured) with PECVD layers only based on a-Si (e.g., a-Si (p), a-Si (n), i-a-Si) without any additional protective layers or back-reflector. The lifetime of such industrially relevant bottom-cells from CSEM is 2-2.5 ms at 5e10¹⁵ cm⁻³. Hence, the optoelectronic performance of 1cm² vs. larger bottom cells is unfortunately different due to unavailability and impracticality. Reference 1 cm² devices fabricated with the industrially relevant CSEM bottom cells on average, provide 31.5% efficiency, which is only 1.4 abs% more than the in-house 60 cm² results (30.1% champion).
11. Moreover, in a 1cm² device, the pFFs of each sub-cell are relatively high and similar (e.g., 85-86%), which makes the optimum current mismatch condition close to current matching. However, for the 60 cm² devices, due to challenges in scaling up the perovskite sub-cell, the pFF of the top-cell is lower than the bottom-cell. Such asymmetry in pFF moves the optimum efficiency point towards a bottom-limited scenario. Hence, we observed higher FF's and PCE's when the bottom-cell limitation increases (e.g., when Jsc of the top-cell increases and Jsc of the bottom-cell decreases). Hence, for 60 cm² devices, we increased the perovskite thickness, as shown in **Fig.R17** from the lower response of the Si sub-cell of the 60 cm² device in the wavelength range 600 nm to 750 nm.
12. Based on a loss-estimation we think that, by utilizing a back-reflector and optimized PECVD films and protective layers, a 60 cm² tandem device with the top cell demonstrated in this work can achieve: VOC= 1.98V, FF = 80%, and JSC = 19.75 mA/cm² – PCE = 31.28%.

Lastly, the number of fabricated 1 cm² devices reaches 250 in this work, however, for the 60 cm² devices, this number is only around 20, and for 4 cm², 14. Hence, the effort demonstrated and the throughput are significantly different. Larger devices are only at the beginning of their learning curve.

#16. The difference in Jsc between the 1 cm² and 60 cm² devices is significantly higher than the Jsc difference caused by shadowing losses. Could the authors provide an analysis of the mechanisms contributing to this Jsc loss?

The EQE of the 1 cm² (measured in between fingers but with shading correction – EQE multiplied by 0.9875) and 60 cm² (measured full area in Fraunhofer ISE Callab), can be seen in Fig.R17. Firstly, the perovskite top-cell of the 60 cm² device is thicker, lowering the 600-750 nm response of the 60 cm² SHJ cell. Increasing C60 parasitic absorption (due to thicker films) is responsible for the decreasing EQE of the 60 cm² perovskite sub-cell in the range of 400 nm to 550 nm. Higher parasitic absorption of the ITO compared to IZrO lowers the response likely over the whole spectrum. Due to the ITO parasitic response, the 60 cm² top-cell response is only slightly higher than 1 cm² response around 700 nm. The IR response of the 1 cm² device is only marginally better, likely due to high-transparency IZrO and SiOx-based reflector, even though 60 cm² champion device is double-sided nanotextured.

The UV (350 nm to 400 nm), response of the cells is similar despite the thicker C60 and less transparent TCO, because 1 cm² cells utilize thicker ALD-SnOx (175 cycles from PICOSUN) compared to 125 cycles from Oxford ALD for the large devices. Thus, major differences causing lower Jsc in the 60 cm² device are the parasitic absorption losses due to ITO and C₆₀, and the thicker perovskite to benefit from higher FF's due to current mismatch. We included the following EQE comparison figure in the SI as Fig.S(30) and added the following sentence to the main text;

“The comparison of the EQE response of a 1 and 60 cm² device can be seen in Fig.S30, highlighting the superior optics of the small area due to a high-transparency top-electrode stack (e.g., thinner C₆₀, thinner and more transparent IZrO instead of ITO).”

Figure R17. EQE of 1 cm² and 60 cm² devices.

Reviewer #3 (Remarks to the Author):

The manuscript presents an important advance in scaling up the perovskite silicon tandem solar cells, achieving 28.9 % certified efficiency over 60 cm² through a homogeneous dual passivation strategy (pFBPA and PCl). The approach combining phosphonic acid surface passivation and piperazinium chloride interface passivation is convincing and well supported by several characterization and DFT calculations. The study is thorough and the results are of high relevance to the field. I recommend this manuscript for publication after the authors address the following clarifications.

We would like to thank the reviewer for the valuable comments and feedback and for pointing out possible concerns. In the following, we addressed these points to the best of our knowledge. Moreover, changes are highlighted in the revised manuscript.

Comments and requests for clarification:

1. The manuscript primarily focuses on steady state PL data. To better understand charge extraction and recombination dynamics at the perovskite/C₆₀ interface, could the authors provide transient PL (TRPL) decay curves for the different passivation conditions, both with and without the C₆₀/SnO_x layers?

We thank the reviewer for the suggestion, and we conducted TRPL measurements on ITO/SAM/perovskite/passivation with and without C₆₀/SnO_x. We included TRPL results in the **Fig.2(b), Fig.S(7-8) and Table S2**. Samples with SP/IP/SIP showed enhanced carrier lifetimes in the presence of ETL-stack, confirming PL-mapping results, where SIP reached the highest lifetime. Moreover, the charge transfer rate is investigated and found that IP provides the fastest transfer, followed by SIP.

Following section is included in the main text in the section related to TRPL results;

“We then measured the dynamics of charge carriers using time-resolved photoluminescence, as shown in Fig.S7. We select the illumination conditions to closely mimic the carrier densities under solar operation and set the laser fluence to generate a carrier density of approximately $2 \times 10^{15} \text{ cm}^{-3}$. The measurements are performed on partial stacks with and without the ETL-stack (C₆₀/SnO_x) and illuminated from the ETL interface to be more sensitive to this interface. In contrast to the control samples with ETL, which introduces an additional recombination channel, we observe an extension of the lifetime when C₆₀ is deposited atop the SP/IP/SIP modified perovskite layers. The highest lifetime (5.3 μs) is achieved with SIP in the presence of the C₆₀/SnO_x, see **Table S2**. This concept is well-known from c-Si research and represents the scenario where a passivating contact improves minority carrier lifetimes³⁵. We observe the same trend as in the PL mapping experiment, **Fig.2(a)**. Furthermore, we analysed the data with C₆₀ in line with Krogmeier *et al.*³⁶ and calculated the differential lifetimes, **Fig.S8(a-b)**. We observe in the differential lifetime plots two regimes, see **Fig.S8(c)**. In the initial phase, charge extraction is witnessed, where the fastest transfer occurs for samples with IP, followed by SIP. In the second phase, the plateauing indicates the minority carrier lifetime, which is highest for SIP in line with the higher V_{oc} in the devices. This means that IP facilitates charge transfer the best, while the combined SIP has slightly slower extraction, but also reduced recombination,

being the overall best condition for device performance.”

2. Regarding stability, could the authors provide statistical data (e.g., number of tested samples, distribution of T_{80} lifetimes) to confirm that the observed degradation trends under operational conditions are representative and reproducible?

To Fig.S(33 and 36), we included the number of devices tested and T_{80} lifetimes. We note that for the control device, since the performance is still increasing for the tested duration, we do not estimate the T_{80} lifetime.

We tested the stability of the single junction devices in different batches, and the accelerated degradation observed with the passivation is repeatably demonstrated. We chose representative samples (i.e., closest to calculated average of multiple cells measured)

Figure R18. Operational stability of single junction devices with different surface treatments

We also tested other interface passivating PAs like BPA and observed similar accelerated degradation, see Fig.R(18 and 19).

Figure R19. Operational stability of control and BPA-treated device.

3. In large-area devices, how does the 130 °C Ag curing process impact the structural or chemical integrity of the passivation layer? Was there any indication of degradation or performance loss associated with this step?

We thank the reviewer for pointing out this important point. We did not specifically investigate the chemical and structural integrity of passivation upon 130 °C stressing. However, the curing process of the Ag paste induces performance losses likely associated with the degradation of the absorber. When the PbX₂-excess amount is kept at 6% for the CsFAMA-TH absorber, which is the baseline for 1 cm² devices, the performance is on average 25%, see Fig.R(20). Upon decreasing the PbX₂-excess amount of the absorber down to 3%, the high performance is preserved for the large area tandems, reaching >29% average performance. Hence, we deem it likely that, rather than passivation, the absorber decomposes due to high PbX₂-content demonstrated also in literature by Tumen-Ulzii et al. and Fu et al. (DOI: 10.1002/adma.201905035 and DOI: 10.1039/C9EE02043H).

Figure.R20. 60cm² tandems with different absorbers (CsFA-TH with no PbX₂-excess, CsFAMA-TH with 3%-excess PbX₂, CsFAMA-TH with 6% PbX₂-excess)

To improve the clarity in the manuscript, we updated **Fig.S(38)** to illustrate the importance of decreasing the PbX₂-content in the absorber to withstand such thermal stress.

4. Is there any experimental evidence (e.g., UPS shifts, XPS coverage estimates) that provides insight into the molecular coverage or uniformity achieved by the pFBPA and PCI treatments?

In principle, in XPS, using a Tougaard approximation of the background, chemical-state analysis in complex multilayer or heterogeneous samples can be enhanced. Especially if angle-resolved XPS measurements (like the ones we demonstrated in this work - DOI:10.1002/adma.202311745) have been done, such analysis can be made easier. However, in this work, we did not conduct AR-XPS, and we also do not have access to the tool (Quases) for such analysis at the moment. We tried to conduct EDX analysis in top-view SEM for such passivation layers atop perovskite. However, it was not possible to distinguish the noise from the actual signal, which resulted in similar maps with and without the presence of passivating layers. In our previous work (DOI:10.1126/science.adg0091) with similar molecules (e.g., pFBPA) utilized, using SIMS-mapping, we managed to map the presence of ¹⁹F⁻ and coverage of such clusters on the surface is estimated as 10%.

5. Please Include the figures of merit (Voc, Jsc and FF) extracted during the ageing studies to complement the efficiency data.

Here, we included the plots of V_{MPP} and J_{MPP} for control and SIP devices during ageing studies in SI as **Fig.S36**. The decrease in J_{MPP} for the SIP devices is attributed to enhanced ionic loss demonstrated by (DOI: 10.1038/s41560-024-01487-w) and proved by the BACE and FS measurements (see Fig.S32-33). Initially V_{MPP} of the SIP devices is more than 100 mV higher due to surface and interface passivation compared to the control device. Whereas, after approximately 50 hours, V_{MPP} of control and SIP devices coincide.

Figure R21. V_{MPP} and J_{MPP} over time during ageing studies for a control and a SIP device.

We thank the reviewers for their valuable comments. We answer their comments in red below.

Reviewer #2 (Remarks to the Author):

The authors had revised the manuscript and answered the comments #1-#10 nicely, I don't have any other comments, however, for #11 and #12 I have two more comments.

#11, The author claims using our equivalent circuit modeling for tandems (if required, we can provide more details). We would appreciate the author giving more details about the modeling and specify the data sources. (Data supported by published literature or industry consensus)

‘Using our equivalent circuit modeling for tandems (if required, we can provide more details), we simulated the influence of increasing the light intensity from 1 sun to 1.016 sun. The changes in the V_{OC} and FF with comparison to the scaling method can be seen in the following table (Table R1);’

Table R1: Equivalent circuit modeling results to compare the effect of JSC scaling to FF, VOC and overall PCE.

Condition	V_{OC} (V)	FF (%)	J_{sc} (mA/cm ²)	PCE (%)
1 – 1 sun	1.960	80.86	19.000	30.110
2 – 1.016 sun	1.961	80.78	19.304	30.579
3 – only Jsc scaled x 1.016	1.960	80.86	19.304	30.594

In the numerical calculations, equivalent circuit (double diode model) modeling, we use the following circuit model and perform the numerical calculations in MATLAB;

Figure R1. A) Equivalent circuit model of the tandem device utilized. B) Parameters utilized in equivalent circuit modeling.

Similar calculations have been done in our previous work using the same model ([DOI: 10.1016/j.joule.2024.04.015](https://doi.org/10.1016/j.joule.2024.04.015)) and a similar model is utilized by Boccard *et al.* ([DOI:10.1021/acseenergylett.0c00156](https://doi.org/10.1021/acseenergylett.0c00156)). Compared to the work by Turkay *et al.*, here we have adjusted the J_{SC-top} , $J_{SC-bottom}$, R_{sh-top} , J_{02} , parameters to relatively match the measured JV parameters (in-house) for the 60 cm² device (see Fig.3(f)). Here, the simulated outcome with certain device parameters can be visualized (Table R2);

Table R2. Simulation results from the equivalent circuit modeling.

J_{sc_top}	J_{sc_bot}	J_{sc}	V_{oc}	V_{oc-top}	V_{oc-bot}	FF	Eff	V_{mpp}	J_{mpp}
20	19	19.002	1960.4	1.2619	0.69851	0.80854	30.119	1622.8	18.56

#12, Did the authors account for potential inconsistencies in testing due to device degradation? If so, please specify the exact data variations caused by degradation. If not, please elaborate on how differences in the test setup (masks, probes, and so on.) may lead to discrepancies in electrical parameters and explain the underlying reasons.

Table R3: Comparison of in-house and certified JV parameters

	V_{oc} (mV)	J_{sc} (mA/cm ²)	FF (%)	PCE
Certified	1965	18.86	78.2	28.98%
In-house	1974	19.18	79.2	29.98%

We thank the Reviewer for the valuable comment. We did take into consideration potential inconsistencies due to degradation, especially in sight of certification. However, we would like to precise that the shelf stability (stored in the dark under N₂ atmosphere) is quite remarkable. Please, see the data for 60 cm² devices – see **Fig.R3**. We updated Fig.S32 to include the shelf stability of large-area devices. The shelf stability of large area devices is relatively good, comparable to 1 cm² devices. This suggests that the devices are quite stable during the waiting time between the in-house characterization and the certification.

However, the certification procedure irradiates the device under maximum power point tracking until the power stabilizes. For our device, this process took approximately 850 s (see **Fig.R4**). Then, the MPP is averaged over the last 300 s to get the certified value. This procedure is significantly longer than our in-house measurements, where we track the MPP for 100 s (see **Fig.S28** – where MPP – 29.8% is already slightly lower (0.2%abs) than champion forward scan – 29.98%), followed by forward and reverse JV scans multiple times until the PCE saturates. Hence, such procedures are relatively different, and cells are more sensitive to MPP tracking than consecutive JV scanning (where we don't observe degradation in performance). In conclusion, it is possible that during the 850 s tracking, under MPP, V_{oc} , and FF might slightly change. Which is in line with the 1%abs FF and 9 mV V_{oc} discrepancy between in-house vs. certification.

Finally, we would like to point out one final difference regarding the J_{sc} reported in house and in the certification. For the in-house measurements, we used a mask with 58.4cm² aperture, while for the

certification the mask was 60.74 cm². Hence, we believe the main J_{SC} difference originates from the differences in masking and the application of the mask on the active area.

We hope these thorough explanations satisfy the Reviewer's request.

Figure R3. Shelf stability of two 60 cm² devices.

Figure R4. Maximum power point tracking during certification by Fraunhofer ISE CalLab.